# Knowledge Composition using Task Vectors with Learned Anisotropic Scaling

**Frederic Z. Zhang**[*]   **Paul Albert**[*]   **Cristian Rodriguez-Opazo**
**Anton van den Hengel**   **Ehsan Abbasnejad**

Australian Institute for Machine Learning    The University of Adelaide

`{firstname.lastname}@adelaide.edu.au`
https://github.com/fredzzhang/atlas

## Abstract

Pre-trained models produce strong generic representations that can be adapted via fine-tuning on specialised datasets. The learned weight difference relative to the pre-trained model, known as a task vector, characterises the direction and stride of fine-tuning that enables the model to capture these specialised representations. The significance of task vectors is such that simple arithmetic operations on them can be used to combine diverse representations from different domains. This paper builds on these properties of task vectors and aims to answer (1) whether components of task vectors, particularly parameter blocks, exhibit similar characteristics, and (2) how such blocks can be used to enhance knowledge composition and transfer. To this end, we introduce aTLAS, an algorithm that linearly combines parameter blocks with different learned coefficients, resulting in anisotropic scaling at the task vector level. We show that such linear combinations explicitly exploit the low intrinsic dimensionality of pre-trained models, with only a few coefficients being the learnable parameters. Furthermore, composition of parameter blocks enables modular learning that effectively leverages the already learned representations, thereby reducing the dependency on large amounts of data. We demonstrate the effectiveness of our method in task arithmetic, few-shot recognition and test-time adaptation, with supervised or unsupervised objectives. In particular, we show that (1) learned anisotropic scaling allows task vectors to be more disentangled, causing less interference in composition; (2) task vector composition excels with scarce or no labelled data and is less prone to domain shift, thus leading to better generalisability; (3) mixing the most informative parameter blocks across different task vectors prior to training can reduce the memory footprint and improve the flexibility of knowledge transfer. Moreover, we show the potential of aTLAS as a parameter-efficient fine-tuning method, particularly with less data, and demonstrate that it can be easily scaled up for higher performance.

## 1 Introduction

One practical advantage of neural networks is the fact that knowledge learned from a previous problem, in the form of network weights, can be transferred to solve other related problems. Commonly referred to as transfer learning [6, 73], this technique is often applied when a model trained on a general-purpose dataset—ImageNet [52] for many years—is fine-tuned on other datasets to improve

---

[*]Equal contribution. Listed order was determined by a coin toss. Fred formalised the idea; developed the original codebase; conducted experiments on task arithmetic; and drafted the paper. Paul designed and conducted experiments on few-shot recognition, test-time adaptation, parameter-efficient fine-tuning; investigated numerous properties of task vector compositions; and was crucially involved in every stage of the work.

38th Conference on Neural Information Processing Systems (NeurIPS 2024).

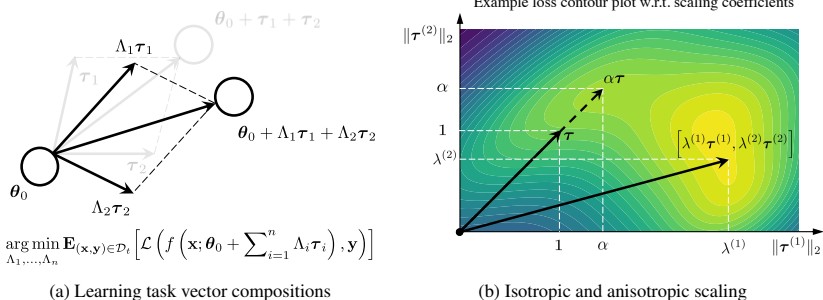

(a) Learning task vector compositions       (b) Isotropic and anisotropic scaling

Figure 1: Illustration of (a) learning task vector compositions ($n = 2$, $\boldsymbol{\theta}_0$ denotes the weights of a pre-trained model) and (b) the flexibility of anisotropic scaling. Assume a task vector $\boldsymbol{\tau} = \left( \boldsymbol{\tau}^{(1)}, \boldsymbol{\tau}^{(2)} \right)$ has two parameter blocks, learning anisotropic scaling grants more flexibility when combining task vectors.

performance on downstream problems. In the past, classification models [18, 53] have been used as the medium for such knowledge transfer, which played a crucial part in the success of detection and segmentation [7, 19, 51, 66–68]. In recent years, foundation models [4] trained on broad data, CLIP [47] particularly, have demonstrated strong performance on a multitude of tasks, even when applied in a zero-shot manner. Besides the conventional way of exploiting the knowledge in these models via fine-tuning, recent works [28, 44, 62] have presented more direct measures to manipulate the network weights. In particular, Ilharco et al. [28] showed that, a task vector, defined as the weight difference between a pre-trained and a fine-tuned model, can be used as a carrier of the task-specific knowledge learned via fine-tuning. As such, multiple task vectors, when combined with simple arithmetic, can form a multi-task model that largely retains its performance across all fine-tuning tasks. Linearisation techniques [44], in addition, have been shown to further enhance this compositionality.

Intrigued by this phenomenon, we investigate the potential of task vectors being knowledge carriers in this paper, by learning linear combinations of them (Figure 1a) for various problems. In particular, parameter blocks, e.g., weights and biases, tend to encode different learned representations in different layers. We thus learn an independent scaling coefficient per block for more precise adjustments tailored to the unique roles of each parameter block. This results in anisotropic scaling of task vectors (Figure 1b), and allows us to exploit their modularity in knowledge composition, granting higher controllability when steering the behaviours of a model for task arithmetic [28].

The potential applications of task vector composition extend beyond model editing. With the coefficients being the only learnable parameters, our method exploits the rich knowledge encapsulated in the task vectors by searching in a low-dimensional coefficient space. As a result, it is a competitive parameter-efficient fine-tuning (PEFT) method, and is particularly effective in cases where labelled data is scarce. This offers new opportunities for few-shot learning [34, 69] and test-time adaptation [35, 57]. Furthermore, for multi-purpose models such as CLIP [47], variants of the model trained with different data sources or fine-tuned on different downstream tasks are often available [26]. These resources constitute a significant knowledge bank, with task vectors being the knowledge carrier. Many learning problems may be simplified to learning a combination of task vectors.

Our primary contribution is a learning algorithm named aTLAS, wherein otherwise complex learning problems can be framed as learning linear combinations of task vectors. The algorithm is broadly applicable to optimising supervised and unsupervised objectives. Its effectiveness is demonstrated in task arithmetic, few-shot recognition, test-time adaptation and parameter-efficient fine-tuning, where we show that (1) learning linear combinations of task vectors directly exploits the low intrinsic dimensionality of pre-trained models [1, 33], resulting in a small number of learnable parameters; (2) standard task vectors, otherwise inferior to linearised variants [44] in task arithmetic, can produce stronger multi-task models with learned anisotropic scaling; (3) aTLAS is effective in low-data regimes, and improves the accuracy of CLIP by 6.5 absolute points averaged over 22 datasets with unlabelled data; (4) aTLAS is complementary to previous few-shot adaptation methods, in that one third of the examples it improves upon are unique; (5) aTLAS as a few-shot learning method is less prone to domain shift, and achieves better generalisation on out-of-domain datasets; (6) the most informative parameter blocks from different task vectors can be mixed prior to training, allowing for flexible and efficient knowledge transfer under memory constraints; (7) aTLAS is a strong PEFT method when data is limited, and existing PEFT methods such as low-rank adaptations (LoRA) [23] can be seamlessly integrated into aTLAS to improve memory efficiency.

## 2 Models and task vectors

As Ilharco et al. [28] demonstrated, task vectors exhibit many intriguing properties across a wide range of models, such as CLIP [47], GPT-2 [46] and T5-based models [48]. To facilitate more in-depth experimentation and analysis, we focus on the CLIP model in this paper, due to its wide availability and manageable size. In particular, we follow previous practice [28, 44] and acquire task vectors by fine-tuning the image encoder, with the text representations frozen. This ensures that image encoders fine-tuned on different datasets produce features residing in the same representation space, through a common text encoder. The task vectors obtained from these fine-tuned encoders can thus be combined more effectively to form a unified multi-task model.

Formally, denote the CLIP image encoder by $f : \mathcal{X} \times \Theta \to \mathcal{Z}$, such that for input image $\mathbf{x} \in \mathcal{X}$ and parameters $\boldsymbol{\theta} \in \Theta$, $\mathbf{z} = f(\mathbf{x}; \boldsymbol{\theta})$ is the learned latent representation for the input image. Denote the weights of a pre-trained model by $\boldsymbol{\theta}_0$, and the weights of its fine-tuned variant by $\boldsymbol{\theta}_i, i \in \mathbb{N}^+$, where $i$ indexes a dataset $\mathcal{D}_i$. We follow Ilharco et al. [28] and define a task vector as $\boldsymbol{\tau}_i = \boldsymbol{\theta}_i - \boldsymbol{\theta}_0$. In addition, we investigate task vectors produced by linearised variants of the image encoder using the first-order Taylor expansion,

$$g(\mathbf{x}; \boldsymbol{\theta}) := f(\mathbf{x}; \boldsymbol{\theta}_0) + (\boldsymbol{\theta} - \boldsymbol{\theta}_0)^{\mathsf{T}} \nabla_{\boldsymbol{\theta}} f(\mathbf{x}; \boldsymbol{\theta}_0). \tag{1}$$

Ortiz-Jiménez et al. [44] showed that, task vectors obtained from fine-tuning the linearised variants have low disentanglement errors, and exhibit strong compositional properties.

## 3 Learning task vector compositions

Parameters in a neural network, depending on the depth of the layer, often have different significance. For instance, early layers in convolutional neural networks [18, 53] are known for extracting generic, low-level features, such as edges, corners, etc., while deeper layers produce features more specific to the task. We recognise the non-uniform impacts parameters at different layers can have, and do not perform isotropic scaling on task vectors. Instead, weights, biases and any other forms of parameterisation, which we collectively refer to as *parameter blocks*, will be scaled independently.

### 3.1 Proposed method: aTLAS

Formally, denote a task vector with $m$ parameter blocks by $\boldsymbol{\tau} = \left(\boldsymbol{\tau}^{(1)}, \dots, \boldsymbol{\tau}^{(m)}\right)$, where each parameter block $\boldsymbol{\tau}^{(j)}$ is vectorised, and round brackets denote column vector concatenation. We learn a block diagonal matrix $\Lambda$, parameterised as

$$\Lambda = \begin{bmatrix} \lambda^{(1)} I^{(1)} & \cdots & \mathbf{0} \\ \vdots & \ddots & \vdots \\ \mathbf{0} & \cdots & \lambda^{(m)} I^{(m)} \end{bmatrix}, \tag{2}$$

where $\lambda^{(j)} \in \mathbb{R}$ is a learnable coefficient; $I^{(j)}$ denotes an identity matrix with its number of columns matching the dimension of $\boldsymbol{\tau}^{(j)}$; and the superscript $j \in \mathbb{N}^+$ indexes a parameter block. This results in anisotropic scaling of a task vector, that is,

$$\Lambda_i \boldsymbol{\tau}_i = \left(\lambda_i^{(1)} \boldsymbol{\tau}_i^{(1)}, \dots, \lambda_i^{(m)} \boldsymbol{\tau}_i^{(m)}\right), \tag{3}$$

where the subscript $i \in \mathbb{N}^+$ indexes a task vector. As such, assuming a supervised objective, finding the optimal composition of task vectors can be defined as the following optimisation problem

$$\underset{\Lambda_1, \dots, \Lambda_n}{\arg\min} \; \mathbf{E}_{(\mathbf{x}, \mathbf{y}) \in \mathcal{D}_t} \left[ \mathcal{L}\left(f(\mathbf{x}; \boldsymbol{\theta}_0 + \textstyle\sum_{i=1}^{n} \Lambda_i \boldsymbol{\tau}_i), \mathbf{y}\right) \right], \tag{4}$$

where $\mathcal{L}$ is the loss function for a target task; $n$ is the number of task vectors; $\mathbf{y}$ is the labels corresponding to inputs $\mathbf{x}$; $\mathcal{D}_t$ denotes a target dataset. The number of learnable parameters, as a result, is precisely $mn$, Let us denote the solution to the aforementioned optimisation problem by $\{\Lambda_i^\star\}_{i=1}^n$. In inference, model $f(\mathbf{x}, \boldsymbol{\theta}_0 + \sum_{i=1}^{n} \Lambda_i^\star \boldsymbol{\tau}_i)$ will be deployed, which incurs no additional computational cost compared to models trained in the conventional way.

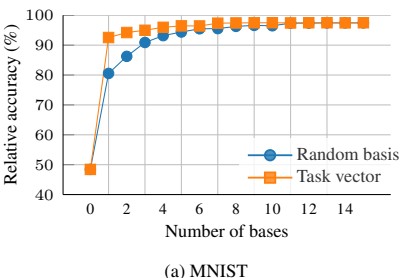
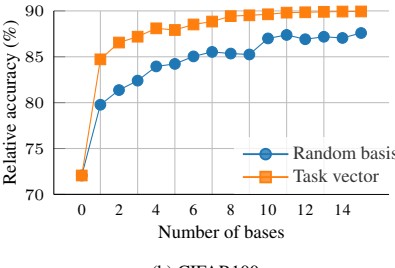

(a) MNIST

(b) CIFAR100

Figure 2: Recognition accuracy versus the number of bases when optimising in a low-dimensional subspace. The accuracy is normalised by that of the fully fine-tuned model. Using task vectors to construct the projection matrix performs consistently better than using random bases on (a) MNIST [32], (b) CIFAR100 [31].

In addition, we investigate the task vectors obtained from fine-tuning linearised variants of the model, i.e., $g(x)$ in Eq. 1. Denote such task vectors by $\widetilde{\tau}$. The learning objective with linearised task vectors can be derived as follows

$$\underset{\Lambda_1,\ldots,\Lambda_n}{\arg\min}\ \mathbf{E}_{(\mathbf{x},\mathbf{y})\in\mathcal{D}_t}\Big[\mathcal{L}\Big(f(\mathbf{x};\boldsymbol{\theta}_0) + (\textstyle\sum_{i=1}^n \Lambda_i\widetilde{\boldsymbol{\tau}}_i)^{\mathsf{T}}\nabla_{\boldsymbol{\theta}}f(\mathbf{x};\boldsymbol{\theta}_0),\mathbf{y}\Big)\Big]. \tag{5}$$

### 3.2 Relation to intrinsic dimensionality

A notable characteristic of aTLAS is its parameter efficiency. To offer more intuitions, we refer to previous findings [1, 33] that deep neural networks often produce solutions residing in a subspace with much lower intrinsic dimensionality. This is measured by finding a minimum number of $d$ parameters, such that learning these parameters ($\hat{\boldsymbol{\theta}} \in \mathbb{R}^d$) leads to approximately the same performance as optimising in the full parameter space ($\boldsymbol{\theta} \in \mathbb{R}^D$). This can be expressed as follows

$$\boldsymbol{\theta} = \boldsymbol{\theta}_0 + P\hat{\boldsymbol{\theta}}, \tag{6}$$

where $\boldsymbol{\theta}_0 \in \mathbb{R}^D$ denotes the pre-trained weights and $P \in \mathbb{R}^{D\times d}$ is a random projection matrix. We demonstrate that learning task vector compositions leads to the same formulation. For brevity of exposition, let us consider compositions at the block level. For the $j$-th parameter block, we have

$$\boldsymbol{\theta}^{(j)} = \boldsymbol{\theta}_0^{(j)} + \sum_{i=1}^n \lambda_i^{(j)}\boldsymbol{\tau}_i^{(j)} \tag{7}$$

$$= \boldsymbol{\theta}_0^{(j)} + \underbrace{\Big[\boldsymbol{\tau}_1^{(j)},\ldots,\boldsymbol{\tau}_n^{(j)}\Big]}_{\text{projection matrix}}\underbrace{\Big[\lambda_1^{(j)},\ldots,\lambda_n^{(j)}\Big]^{\mathsf{T}}}_{\text{learnable parameters}}. \tag{8}$$

We draw a parallel between Eqs. 6 and 8 and note that aTLAS explicitly exploits the low intrinsic dimensionality by learning a small set of coefficients. The number of task vectors, i.e., $n$, is much smaller than the dimension of weight vector $\boldsymbol{\theta}_i^{(j)}$, and is analogous to the intrinsic dimensionality $d$. However, as opposed to using a random projection matrix $P$, aTLAS constructs the projection matrix from task vectors, making use of the learned representations. To demonstrate its advantage, we use the same number of bases for task vectors[2] and random bases[3], and show that task vectors consistently achieve higher performance in Figure 2. These results solidify our understanding of task vectors being knowledge carriers. We thus set out to apply aTLAS to various applications.

## 4 Task arithmetic

Task arithmetic [28] is comprised of a few tasks aimed at editing pre-trained models using task vectors. Following previous practice [28, 44], we conduct experiments under the settings of task negation and task addition on eight image classification datasets (details included in Appendix A).

---

[2]A fixed number of task vectors are selected based on the blockwise gradient. Details can be found in Section 5.2 and Appendix D.6.

[3]Each random basis of the projection is drawn from a Gaussian distribution with the mean and standard deviation to match those of the pre-trained weights in the corresponding parameter block, i.e., $\boldsymbol{\theta}_0^{(j)}$.

Table 1: Performance of task negation averaged across eight datasets. Selected results must maintain at least 95% of the pre-trained accuracy on the control dataset, following previous practice [44]. Best performance in each section is highlighted in bold. Task vector is abbreviated as t.v. Results for each dataset are available in Table 7.

| T.V. | Methods | Models | ViT-B/32 | | ViT-B/16 | | ViT-L/14 | |
| --- | --- | --- | --- | --- | --- | --- | --- | --- |
| | | | Target ($\downarrow$) | Control ($\uparrow$) | Target ($\downarrow$) | Control ($\uparrow$) | Target ($\downarrow$) | Control ($\uparrow$) |
| n/a | Pre-trained | $f(\mathbf{x}; \theta_0)$ | 48.14 | 63.35 | 55.48 | 68.33 | 64.89 | 75.54 |
| Std. | Search | $f(\mathbf{x}; \theta_0 + \alpha\boldsymbol{\tau})$ | 23.22 | 60.71 | 19.38 | 64.66 | 19.15 | 72.05 |
| Std. | aTLAS (ours) | $f(\mathbf{x}; \theta_0 + \Lambda\boldsymbol{\tau})$ | **18.76** | **61.21** | **17.34** | **65.84** | **17.75** | **73.28** |
| Lin. | Search | $g(\mathbf{x}; \theta_0 + \alpha\widetilde{\boldsymbol{\tau}})$ | 11.54 | 60.74 | 10.88 | 65.54 | 12.78 | 72.95 |
| Lin. | aTLAS (ours) | $g(\mathbf{x}; \theta_0 + \Lambda\widetilde{\boldsymbol{\tau}})$ | **11.06** | **61.02** | **10.16** | **65.58** | **12.61** | **73.14** |

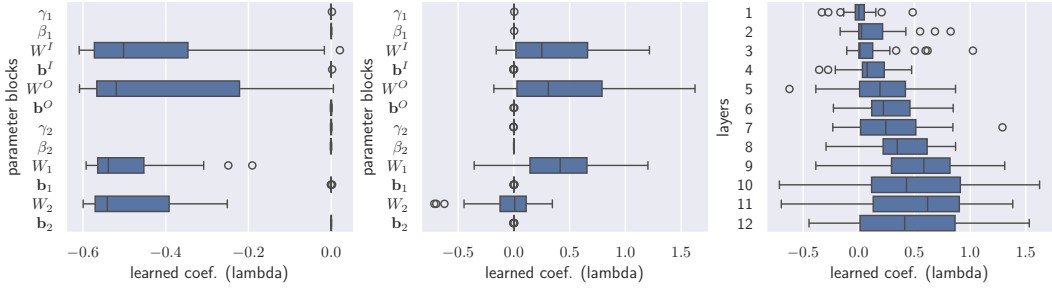

(a) Coef. of param. blocks (negation)    (b) Coef. of param. blocks (addition)    (c) Coef. by layer/depth (addition)

Figure 3: Box-and-whisker plots for the learned coefficients. As each transformer layer consists of a fixed set of parameter blocks, we visualise the distribution of coefficients for these parameter blocks across all layers, for (a) task negation and (b) task addition, as well as (c) distribution of coefficients by layer. We denote the learnable LayerNorm parameters by $\gamma$ and $\beta$. Weights and biases are denoted by $W$ and $\mathbf{b}$, respective, with attention layer parameters indexed by superscripts and the MLP parameters indexed by subscripts.

Previous works acquire the optimal isotropic scaling factor on task vectors via a hyper-parameter search on validation sets. As such, we learn anisotropic scaling matrices on the same validation sets, and visualise the learned coefficients to shed light on this mechanism.

## 4.1 Task negation

Task negation aims to reduce undesired biases, characterised by the performance, on a target task, while maintaining performance on a control dataset, ImageNet [52] in this case. Denote the validation sets for the target and control tasks by $\mathcal{D}_t$ and $\mathcal{D}_c$, respectively. We perform a simultaneous gradient ascent on the target task and gradient descent on the control task, described as follows,

$$\underset{\Lambda_t}{\arg\min} \, \mathbf{E}_{(\mathbf{x},\mathbf{y})\in\mathcal{D}_t}[-\mathcal{L}(f(\mathbf{x}; \boldsymbol{\theta}_0 + \Lambda_t\boldsymbol{\tau}_t), \mathbf{y})] + \mathbf{E}_{(\mathbf{x},\mathbf{y})\in\mathcal{D}_c}[\mathcal{L}(f(\mathbf{x}; \boldsymbol{\theta}_0 + \Lambda_t\boldsymbol{\tau}_t), \mathbf{y})], \quad (9)$$

where $\boldsymbol{\tau}_t$ is the task vector for the target dataset, and cross-entropy loss is used. The learning objectives with linearised task vectors can be derived easily based on Eq. 5, and so are omitted.

We summarise the task negation results in Table 1, and show that our method significantly improves upon standard task vectors, while the improvement upon linear task vectors is less prominent. In particular, we observe that weights matrices tend to have much larger negative coefficients, as shown in Figure 3a. To investigate this, we instead only learn coefficients for the weight matrices, with zero coefficients on other parameter blocks, effectively reducing the number of learnable parameters by two thirds. With ViT-B/32 as the backbone, we observe an average accuracy of 20.14 (vs. 18.76) on target tasks and 61.23 (vs. 61.21) on the control task, which shows that weight matrices carry majority of the knowledge required for task negation.

## 4.2 Task addition

Task addition aims at producing a multi-task model using task vectors acquired from a range of datasets. We utilise task vectors from the eight image classification datasets, and learn the anisotropic

Table 2: Performance of task addition averaged across eight datasets. We report the absolute accuracy (Abs.) and the relative accuracy (Rel.) with respect to the fine-tuned model. Best performance in each section is highlighted in bold. Task vector is abbreviated as t.v. Results for each dataset are available in Table 8.

| T.V. | Methods | Models | ViT-B/32 | | ViT-B/16 | | ViT-L/14 | |
|---|---|---|---|---|---|---|---|---|
| | | | Abs. (↑) | Rel. (↑) | Abs. (↑) | Rel. (↑) | Abs. (↑) | Rel. (↑) |
| n/a | Pre-trained | $f(\mathbf{x};\theta_0)$ | 48.14 | - | 55.48 | - | 64.89 | - |
| Std. | Search | $f\big(\mathbf{x};\theta_0 + \alpha\sum_i \boldsymbol{\tau}_i\big)$ | 70.12 | 77.24 | 73.63 | 79.85 | 82.93 | 87.92 |
| | aTLAS (ours) | $f\big(\mathbf{x};\theta_0 + \sum_i \Lambda_i \boldsymbol{\tau}_i\big)$ | **84.98** | **93.79** | **86.08** | **93.44** | **91.36** | **97.07** |
| Lin. | Search | $g\big(\mathbf{x};\theta_0 + \alpha\sum_i \widetilde{\boldsymbol{\tau}}_i\big)$ | 74.67 | 85.17 | 77.51 | 86.21 | 84.75 | 91.86 |
| | aTLAS (ours) | $g\big(\mathbf{x};\theta_0 + \sum_i \Lambda_i \widetilde{\boldsymbol{\tau}}_i\big)$ | **83.42** | **95.42** | **85.38** | **95.10** | **88.65** | **96.12** |

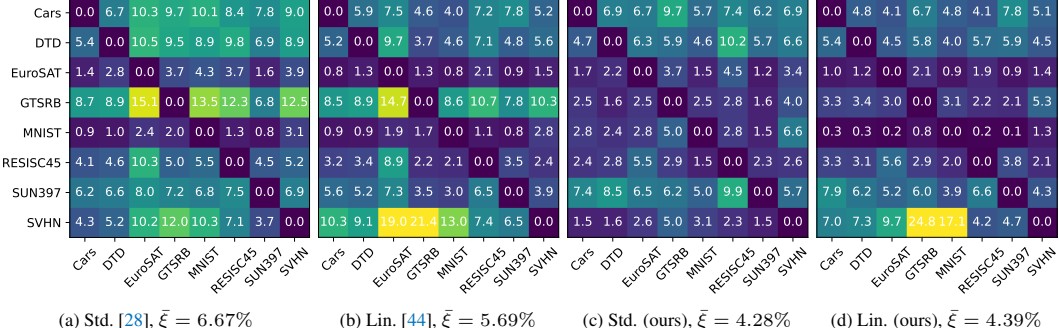

(a) Std. [28], $\bar{\xi} = 6.67\%$  (b) Lin. [44], $\bar{\xi} = 5.69\%$  (c) Std. (ours), $\bar{\xi} = 4.28\%$  (d) Lin. (ours), $\bar{\xi} = 4.39\%$

Figure 4: Disentanglement errors between each pair of datasets. Each row reflects the percentage of data in the corresponding dataset that have altered predictions after combining two task vectors. Our method achieves stronger task addition performance as a result of less interference amongst task vectors.

scaling matrices with the objectives described in Eqs. 4, 5 using the cross-entropy loss. The training data is comprised of the validation sets for all eight dataset, i.e., $\mathcal{D}_t = \bigcup_{i=1}^{8} \mathcal{D}_i$.

Performance comparison against previous methods is shown in Table 2, where our method yields substantial improvements. Interestingly, we note that with previous methods [28, 44], linear task vectors outperform the standard ones in terms of absolute accuracy, while the converse is true with our method. To investigate this, we compute the pairwise disentanglement error $\xi$ [44], which measures the percentage of data with inconsistent predictions when two task vectors are combined (more details in Appendix C.2). Results in Figure 4 show that standard task vectors with learned anisotropic scaling achieve the lowest average error, indicating less interference in task vector composition. Along with higher fine-tuning accuracy, previously referred to as the non-linear advantage [44], standard task vectors demonstrate stronger performance in task addition.

Furthermore, we again observe that weight matrices have consistently larger coefficients in Figure 3b, and learning coefficients on weight matrices alone results in an accuracy of 84.17 (vs. 84.98) using ViT-B/32. This suggests that weight matrices in transformers are the primary knowledge carrier, which enabled knowledge composition and negation. Note that for better clarity in visualisation, we add $L_1$ regularisation on the learned coefficients during learning, which causes marginal performance drop (84.23 vs. 84.98) but significantly improves interpretability. In addition, we observe substantially higher coefficients on deeper layers (Figure 3c). This aligns with our understanding that early layers extract generic features that do not vary significantly across datasets [29], while the deeper layers produce task-specific features and require more careful adaptations.

## 5 Knowledge transfer in low-data regimes

Beyond model editing for task arithmetic, we explore the idea of transferring existing knowledge in task vectors to previously unseen tasks. To this end, we use the CLIP [47] model and a total of 22 image classification datasets, each of which produces a task vector. We defer the details of datasets and the process to acquire task vectors to Appendix A. Denote the set of available task vectors by $T = \{\boldsymbol{\tau}_i\}_{i=1}^{n}$, and the dataset corresponding to task vector $\boldsymbol{\tau}_i$ by $\mathcal{D}_i$. For each target dataset $\mathcal{D}_t$, we

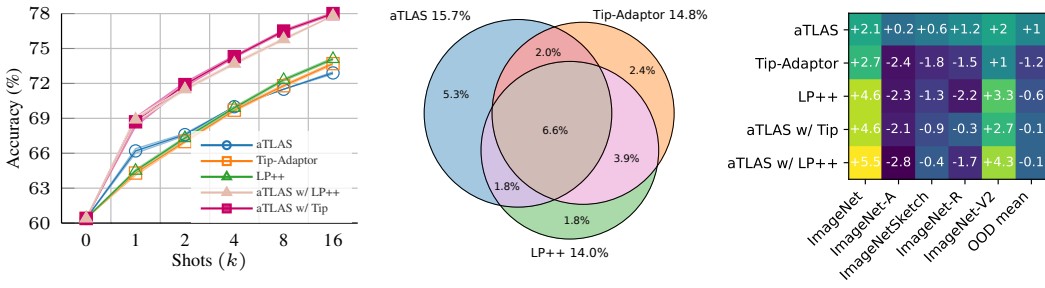

(a) Few-shot recognition performance     (b) Images (%) that become correctly classified     (c) Improvements on OOD datasets

Figure 5: Few-shot experiment results averaged across 22 datasets and three seeds, showing (a) comparison against state-of-the-art few-shot methods with ViT-B/32 backbone and (b) percentage of images in the validation sets that become correctly classified after applying few-shot methods. We also show (c) performance difference compared to pre-trained CLIP model on OOD datasets. More detailed results are included in Appendix D.

learn task vector compositions using the subset $T \setminus \{\boldsymbol{\tau}_t\}$, excluding the task vector for the target dataset to avoid information leakage. We test our method in few-shot and test-time adaptation, to demonstrate its effectiveness in low-data regimes. Notably, we observe that task vectors complement existing few-shot methods. Combining aTLAS with them thus leads to significant improvements.

## 5.1 Few-shot adaptation

Few-shot recognition requires learning new objects or concepts using a limited amount labelled data—$k$ per class for $k$-shot. Following previous practice [69], we approach this problem by adapting a pre-trained CLIP model [47] to each target dataset $\mathcal{D}_t$. We use the subset of task vectors $T \setminus \{\boldsymbol{\tau}_t\}$ and $k \in \{1, 2, 4, 8, 16\}$ images from dataset $\mathcal{D}_t$. During training, we adopt the cross-entropy loss and minimise objectives described in Eqs. 4 and 5 for standard and linear task vectors, respectively.

We compare against Tip-Adapter [69] and LP++ [25] using CLIP with ViT-B/32 backbone, across 22 datasets over three random seeds, and summarise the results in Figure 5a. We show that with $k = 1$, our approach, aTLAS, significantly outperforms previous methods, demonstrating the effectiveness of knowledge transfer with scarce labelled data. More importantly, we note that the idea of task vector composition is highly complementary to those presented in previous methods. As such, combining aTLAS with them results in significant improvements. This is also illustrated in Figure 5b as a Venn diagram, where we show the percentage of examples in the validation set that are incorrectly classified by the pre-trained model but correctly classified with few-shot methods. Out of the examples aTLAS improves upon, around half are unique compared against either Tip-Adapter or LP++, demonstrating its complementarity. We also found that standard task vectors generally perform better than their linearised counterparts, and so defer the results of linear task vectors to Appendix D.2.

In addition, due to the low number of learnable parameters, aTLAS exhibits strong generalisability. To demonstrate this, we learn task vector composition on ImageNet [52], and test it on out-of-domain (OOD) datasets: ImageNet-A [22], ImageNet-R [21], ImageNet-sketch [60] and ImageNetV2 [50]. We summarise the results in Figure 5c, which shows the performance difference against the pre-trained model. Notably, aTLAS is the only method that consistently improves upon the pre-trained model on OOD datasets, and combining aTLAS with other methods can improve their generalisability.

We also test our method and variants integrated with Tip-Adapter and LP++ using other backbones, including ViT-$\{B/16, L/14\}$ and ResNet-$\{50, 101\}$, and find that the results are consistent with those for ViT-B/32. More details can be found in Appendix D.3.

## 5.2 Task vector budget and selection

In practical applications, there may only be a limited number of task vectors available, or the number of task vectors used in training may be restricted due to memory constraints. To this end, we study the influence of task vector budget $b$ on few-shot recognition performance. We experiment with four selection strategies: (1) random selection; (2) feature-based selection; (3) gradient-based selection; and (4) blockwise gradient-based selection. To elaborate, feature-based selection computes the mean image feature representation of each dataset, and selects $b$ task vectors from datasets most similar

Table 3: Test-time adaptation accuracy averaged over 22 dataset, with $\times 1$ standard error over 3 random seeds. LN refers to tuning the LayerNorm layers. CLIP with the ViT-B/32 backbone is used. Highest performance is highlighted in bold.

| Method | Zero-shot | Contrastive (SimCLR) | | Entropy (SAR) | | Pseudo labelling (UFM) | |
|---|---|---|---|---|---|---|---|
| | | LN | aTLAS | LN | aTLAS | LN | aTLAS |
| Accuracy | 60.4 | $60.4 \pm 0.0$ | $62.7 \pm 0.1$ | $61.2 \pm 0.1$ | $62.9 \pm 0.0$ | $62.2 \pm 0.1$ | $\mathbf{66.9} \pm 0.1$ |

to the target dataset. Gradient-based selection computes the gradient with respect to each of the learnable coefficients, and either select entire task vectors with the highest $L_1$ gradient norm, or select task vectors with the highest blockwise gradient for the corresponding parameter block, and repeat the process for all parameter blocks. The blockwise selection therefore allows parameter blocks across different task vectors to be mixed prior to training. More details can be found in Appendix D.6.

For a task vector budget $b \in \{1, 2, 5, 10, 15, 21\}$, we summarise the few-shot recognition performance in Figure 6. First, we note that the accuracy of aTLAS does not plateau with the maximum number of task vectors available (21), indicating that more task vectors could be beneficial. Second, we find that selecting task vectors based on feature similarity is a simple yet effective approach with sufficient budgets ($b > 5$). Selecting whole task vectors with gradient is less effective, generally on par with random selection. Nevertheless, the blockwise variant achieves the best accuracy, particularly for very low budgets ($b \in \{1, 2\}$), as it is able to exploit knowledge from more task vectors than the budget dictates. We thus deduce that parameter blocks can function as knowledge carriers in isolation, independent of the task vectors to which they belong. In fact, a parameter block $\boldsymbol{\tau}^{(1)}$ as part of the task vector $\boldsymbol{\tau} = (\boldsymbol{\tau}^{(1)}, \ldots, \boldsymbol{\tau}^{(m)})$ can be considered as a task vector by itself, i.e., $(\boldsymbol{\tau}^{(1)}, \mathbf{0}, \ldots, \mathbf{0})$. This modular nature underscores the potential of task vectors for flexible and efficient knowledge transfer.

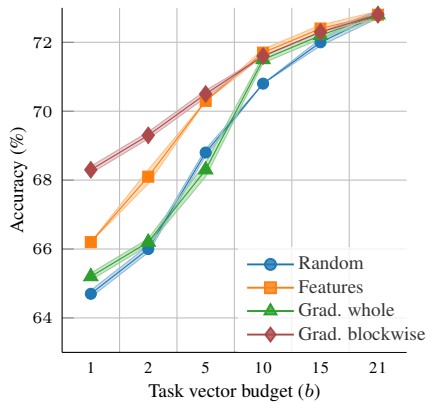

Figure 6: Few-shot performance of aTLAS with various task vector budgets. The accuracy is averaged across 22 datasets and over three random seeds. Standard deviation $\times 1$ is overlaid as the error margin. Performance under the 16-shot setting is visualised, while additional detailed results are included in Table 14.

## 5.3 Test-time adaptation

Test-time adaptation (TTA) [35, 57, 59] assumes no labelled data is available for the target task, requiring the model to adapt in an unsupervised fashion. We conduct experiments under the offline adaptation setting, which allows access to the target dataset. We consider three categories of self-supervised techniques for TTA: constrastive objectives, entropy objectives and pseudo labelling. Contrastive objectives align representations of the same image under different data augmentations. For this category, we adopt SimCLR [9], a simple yet effective method. Entropy objectives encourage the pre-trained model to produce confident predictions on unseen datasets by minimising the entropy over the predictions. This technique was previously explored by Yang et al. [65] in model merging. While effective in simpler cases, it can lead to catastrophic collapse in TTA. Therefore, we utilise a state-of-the-art sharpness-aware entropy minimisation algorithm named SAR [43]. Last, we experiment with an unsupervised pseudo-labelling algorithm inspired by FixMatch [54], which we refer as unsupervised FixMatch (UFM). UFM selects an equal number of highly confident examples per class as the labelled set, and then employs FixMatch to produce pseudo-labels from rest of the unlabelled examples. Details are available in Appendix E.

We summarise the results in Table 3 and compare our method, i.e., learning task vector compositions, against the conventional approach of tuning the layer normalisation parameters [43, 57, 59]. We show that under all self-supervised objectives, aTLAS achieves higher accuracy than tuning the LayerNorm. In particular, LayerNorm has 30k learnable parameters with ViT-B/32 while our method only has 3.5k learnable parameters. We note that with the UFM objective, aTLAS performs the best and improves the accuracy by an average of 6.5 absolute points over the zero-shot baseline.

Table 4: Few-shot recognition performance using standard task vectors or LoRAs as sparse task vectors. Results are averaged across 22 datasets over three seeds, with $\times 1$ standard deviation. The memory consumption for ViT-B/32 backbone is annotated under each variant. For standard task vectors, we learn compositions on all parameter blocks or weight matrices only. For LoRAs as task vectors, we report results with rank 4, 16 and 64.

| Shots ($k$) | Standard task vectors | | LoRAs as task vectors | | |
| --- | --- | --- | --- | --- | --- |
| | All parameter blocks (10.7 GB) | Weight matrices (10.5 GB) | Rank=4 (3.3 GB) | Rank=16 (3.4 GB) | Rank=64 (4.1 GB) |
| 1 | $66.0 \pm 0.2$ | $66.0 \pm 0.1$ | $64.4 \pm 0.1$ | $64.6 \pm 0.1$ | $65.4 \pm 0.1$ |
| 2 | $67.7 \pm 0.1$ | $67.0 \pm 0.2$ | $65.7 \pm 0.0$ | $66.6 \pm 0.2$ | $67.4 \pm 0.1$ |
| 4 | $70.0 \pm 0.0$ | $69.4 \pm 0.2$ | $68.2 \pm 0.0$ | $68.7 \pm 0.1$ | $69.5 \pm 0.2$ |
| 8 | $71.3 \pm 0.1$ | $70.9 \pm 0.0$ | $70.2 \pm 0.2$ | $70.4 \pm 0.1$ | $70.9 \pm 0.1$ |
| 16 | $72.8 \pm 0.1$ | $72.3 \pm 0.0$ | $71.7 \pm 0.1$ | $71.8 \pm 0.1$ | $72.0 \pm 0.1$ |

# 6 Relation to parameter-efficient fine-tuning

One of the key advantages of aTLAS is its ability to adapt pre-trained models with few learnable parameters, making it suitable for parameter-efficient fine-tuning (PEFT). Similar to popular PEFT methods such as low-rank adaptation (LoRA) [23], our approach does not introduce additional modules, thereby avoiding an increase in inference complexity. In addition, since only the encoded weight matrices in LoRAs have non-zero weight difference, LoRAs are in fact sparse task vectors. They can thus be seamlessly integrated into our method, significantly reducing the memory cost.

## 6.1 LoRAs as task vectors

Due to the sparsity and rank deficiency, LoRAs as task vectors may have limited representation capacity and carry less knowledge. Therefore, they may be inferior to standard task vectors for knowledge transfer. We investigate this by learning linear combinations of LoRAs[4] using our method, under the settings of few-shot recognition. Results are summarised in Table 4. We first shed light on the impact of sparsity, and compare two variants of our method that either learns linear combinations of all parameter blocks or just the weight matrices. Results show that sparsity results in an accuracy decrease of around $0.5\%$ on average, except for the one-shot setting. The rank deficiency, on the other hand, causes more substantial accuracy drop. Nevertheless, this can be largely mitigated by increasing the rank. Using a rank of $64$ leads to similar performance compared to learning compositions of only weight matrices in standard task vectors. In conclusion, while the sparsity and rank deficiency introduce some performance drops, especially in low-shot settings, LoRAs are competitive alternatives to standard task vectors due to their low memory cost.

## 6.2 Scalability of aTLAS

Despite the parameter efficiency of aTLAS, its performance is not as competitive when sufficient training data is available. To address this, we devise a strategy to flexibly scale up the number of learnable parameters as needed. Specifically, we randomly divide each parameter block into $K$ partitions, and assign a learnable coefficient to each partition, naturally increasing the number of learnable parameters by $K$-fold. We denote these variants by aTLAS $\times K$. We conduct experiments with these variants using $\{1, 5, 10, 25, 35, 50, 100\}\%$ of the total available training data across the 22 datasets used in Section 5. The results are summarised in Figure 7, showing that our method consistently improves as $K$ increases. Compared to LoRAs, particularly with limited training data,

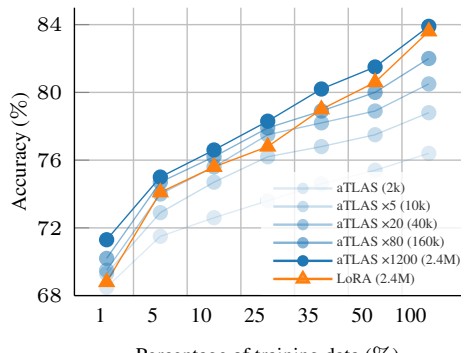

Figure 7: Scalability of aTLAS. We compare the accuracy of our method against LoRAs, and vary the amount of training data. Results are averaged over 22 datasets. Detailed results are included in Table 17.

---

[4]Details about the process to acquire LoRAs are included in Appendix D.7.

our method achieves higher performance with fewer learnable parameters. With sufficient training data, the variant aTLAS $\times 1200$ leads to higher performance with a similar number of learnable parameters, as it is able to exploit the knowledge contained in the task vectors that may otherwise be unobtainable from the target dataset.

# 7 Related work

**Task vectors and model compositions.** Recent studies have demonstrated the possibility of manipulating the behaviours of neural networks directly in the weight space [27, 62, 64]. In particular, task vectors [28], as a carrier of the domain-specific knowledge learned through fine-tuning, exhibit strong compositional properties. Such compositionality can be enhanced via linearisation using first-order Taylor expansion [44], and improves model editing with simple arithmetic, e.g., addition, negation, etc. Yang et al. [65] also investigated the idea of learning layer-wise coefficients to improve task arithmetic. In addition, low-rank adaptations [23], as special forms of task vectors, were shown to also support such arithmetic operations. A recent study [3] also investigated the idea of learning combinations of LoRAs for few-shot recognition.

**Model-based transfer learning.** One interpretation of transfer learning [73] is to exploit the knowledge encapsulated in a pre-trained model for a target domain. Amongst various sub-modules of a pre-trained model, transferring the feature extractor is the most extensively studied. This ranges from early convolutional neural networks [18, 53] to modern transformers [58], from vision backbones [14, 37] to language models [13, 46]. For vision applications, classification models trained on ImageNet [52] have been used as the medium for knowledge transfer. In recent years, contrastively pre-trained multi-modal models such as CLIP [47] have emerged as a prevalent choice. Such models are trained on large volumes of data by aligning image and language representations, leading to strong baselines well suited for transfer learning. CLIP representations have since been use for medical imaging [70], semantic segmentation [72], satellite imaging [40], etc.

**Model adaptation in low-data regimes.** The performance of pre-trained models is often constrained when applied to specific tasks with limited labelled data. To address this limitation, extensive research has been conducted on few-shot adaptation of CLIP [47]. These studies focus on various techniques, including prompt engineering [71], feature adaptation [16], and more recently classifier adaptation [25, 69]. In addition to few-shot adaptation, test-time adaptation represents an even more challenging scenario where no annotated data is available. This typically requires leveraging self-supervised objectives to adapt the model, employing methods such as entropy minimisation [35, 43, 59], contrastive learning [8], pseudo labelling [35] and image rotation prediction [57].

# 8 Conclusion

In this paper, we introduced aTLAS, a learning algorithm that leverages the rich knowledge encapsulated in task vectors through learned linear combinations with anisotropic scaling. Unlike conventional methods that learn network parameters, our approach focuses on learning coefficients on task vectors, significantly reducing the number of learnable parameters. We conducted experiments across task arithmetic, few-shot recognition, test-time adaptation and parameter-efficient fine-tuning, demonstrating the effectiveness of our method with supervised and unsupervised objectives. In particular, we highlighted several properties of aTLAS, including low disentanglement error, robustness against domain shift, effectiveness in low-data regimes, complementarity with existing few-shot methods, etc. These properties paved the way for efficient knowledge composition and transfer.

**Limitations.** As a task vector is defined with respect to a specific pre-trained model, knowledge composition and transfer are not yet feasible across different architectures. This may become possible with suitable projections and remains part of the future work. In addition, combining large numbers of task vectors can consume a substantial amount of GPU memory when training larger models. This can be mitigated by selecting a subset of task vectors, using LoRAs as task vectors or by offloading the computation of task vector composition to CPU, at the cost of training speed decrease. It is also possible to perform task vector composition at bit-width lower than floating point precision, e.g., 4-bit. Similar features are being tested with popular deep learning frameworks such as PyTorch, and we expect the memory requirement of larger models to be less of a constraint in the future.

**Acknowledgements.** This research is funded in part by the Australian Government through the Australian Research Council (Project DP240103278), and the Centre of Augmented Reasoning at the Australian Institute for Machine Learning, established by a grant from the Department of Education. We would like to thank Stephen Gould for his valuable feedback on the paper.

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

# A  Datasets and task vectors

We acquire task vectors by fine-tuning CLIP [47] on a variety of 22 image recognition datasets: (1) Stanford Cars [30], (2) DTD [11], (3) EuroSAT [20], (4) GTSRB [56], (5) MNIST [32], (6) RESISC45 [10], (7) SUN397 [63], (8) SVHN [41], (9) CIFAR10 [31], (10) CIFAR100 [31], (11) ImageNet [52], (12) STL10 [12], (13) Food101 [5], (14) Caltech101 [34], (15) Caltech256 [17], (16) FGVCAircraft [39], (17) Flowers102 [42], (18) Oxford Pets [45], (19) CUB200 [61], (20) PascalVOC [15], (21) Country211 [47], and (22) UCF101 [55]. Fine-tuning was conducted using AdamW optimiser [38], with a learning rate of $10^{-5}$, batch size of 128 and weight decay of 0.1. Details of the datasets, additional dataset-specific hyper-parameters, and the accuracy after fine-tuning for an assortment of backbones are shown in Table 5. We use the same hyper-parameters for the linearised variants of the model.

Table 5: Details of the 22 image classification datasets used in experiments, the number of epochs for fine-tuning and the final accuracy for different backbones of the CLIP model.

| #    | Datasets     | Classes | Splits | | | Epochs | Fine-tuned accuracy | | | | |
|------|--------------|---------|--------|-----|------|--------|------|-------|---------|---------|---------|
|      |              |         | train  | val | test |        | RN50 | RN101 | ViT-B/32 | ViT-B/16 | ViT-L/14 |
| (1)  | Cars         | 196     | 7,330     | 814    | 8,041  | 35 | 61.92 | 68.41 | 78.26 | 84.14 | 91.67 |
| (2)  | DTD          | 47      | 3,384     | 376    | 1,880  | 76 | 73.14 | 72.50 | 78.94 | 81.91 | 84.73 |
| (3)  | EuroSAT      | 10      | 21,600    | 2,700  | 2,700  | 12 | 98.11 | 98.07 | 98.89 | 98.93 | 99.81 |
| (4)  | GTSRB        | 43      | 23,976    | 2,664  | 12,630 | 11 | 97.33 | 97.51 | 99.14 | 98.84 | 99.30 |
| (5)  | MNIST        | 10      | 55,000    | 5,000  | 10,000 | 5  | 99.62 | 99.45 | 99.65 | 99.69 | 99.77 |
| (6)  | RESISC45     | 45      | 17,010    | 1,890  | 6,300  | 15 | 93.16 | 93.27 | 95.94 | 96.59 | 97.14 |
| (7)  | SUN397       | 397     | 17,865    | 1,985  | 19,850 | 14 | 69.65 | 72.26 | 75.40 | 78.12 | 81.98 |
| (8)  | SVHN         | 10      | 68,257    | 5,000  | 26,032 | 4  | 94.30 | 94.58 | 97.38 | 97.70 | 97.97 |
| (9)  | CIFAR10      | 10      | 45,000    | 5,000  | 10,000 | 5  | 93.55 | 95.43 | 98.05 | 98.54 | 99.22 |
| (10) | CIFAR100     | 100     | 45,000    | 5,000  | 10,000 | 6  | 77.55 | 80.15 | 89.09 | 89.95 | 93.01 |
| (11) | ImageNet     | 1,000   | 1,276,167 | 5,000  | 50,000 | 10 | 76.01 | 78.19 | 76.41 | 81.33 | 85.52 |
| (12) | STL10        | 10      | 4,500     | 500    | 8,000  | 4  | 90.15 | 91.55 | 98.55 | 99.20 | 99.62 |
| (13) | Food101      | 101     | 70,750    | 5,000  | 25,250 | 15 | 85.14 | 87.22 | 88.68 | 92.85 | 95.37 |
| (14) | Caltech101   | 101     | 6,941     | 694    | 1,736  | 10 | 87.62 | 85.89 | 94.41 | 95.22 | 94.82 |
| (15) | Caltech256   | 257     | 22,037    | 2,448  | 6,122  | 8  | 88.29 | 90.54 | 92.60 | 94.58 | 97.17 |
| (16) | FGVCAircraft | 100     | 3,334     | 3,333  | 3,333  | 60 | 23.88 | 26.91 | 40.65 | 47.28 | 68.11 |
| (17) | Flowers102   | 102     | 1,020     | 1,020  | 6,149  | 40 | 60.79 | 55.47 | 90.08 | 94.67 | 97.84 |
| (18) | OxfordIIITPet| 37      | 3,312     | 368    | 3,669  | 5  | 75.14 | 77.49 | 92.15 | 93.59 | 95.91 |
| (19) | CUB200       | 200     | 5,395     | 599    | 5,794  | 20 | 58.11 | 59.56 | 73.56 | 77.37 | 86.35 |
| (20) | PascalVOC    | 20      | 7,844     | 7,818  | 14,976 | 10 | 74.88 | 76.87 | 88.42 | 90.35 | 92.05 |
| (21) | Country211   | 211     | 31,650    | 10,550 | 21,100 | 15 | 19.24 | 20.60 | 21.99 | 27.64 | 38.06 |
| (22) | UCF101       | 101     | 7,639     | 1,898  | 3,783  | 20 | 81.63 | 83.00 | 85.01 | 89.14 | 92.55 |

To shed light on the semantic relationships amongst datasets, we extract the features of all images for each dataset, and visualise the distributions as ellipses (Figure 8). Specifically, for each dataset, the mean $\mu_t \in \mathbb{R}^d$ and covariance $\Sigma_t \in \mathbb{R}^{d \times d}$ of image features are computed. Principal component analysis (PCA) is used produce a projection matrix $P \in \mathbb{R}^{d \times 2}$ from the mean features $\mu_t$. Subsequently, the mean and covariance with reduced dimensionality can be expressed as $P^\mathsf{T} \mu_t$ and $P^\mathsf{T} \Sigma_t P$, respectively.

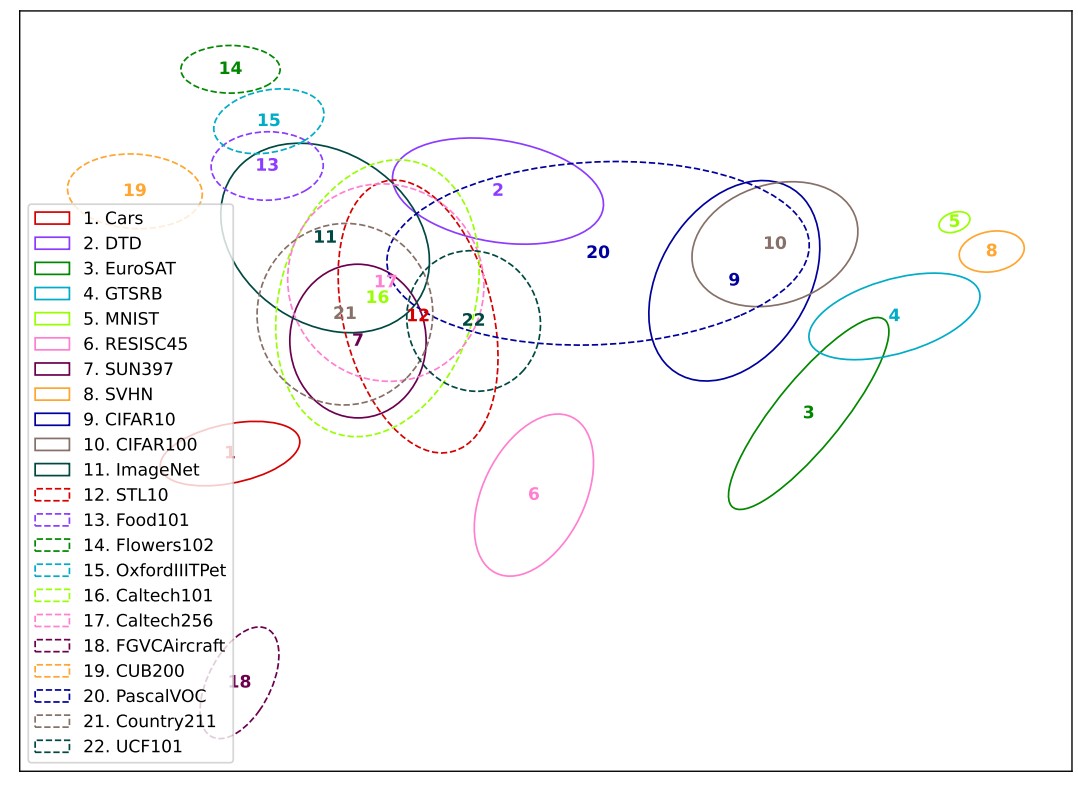

(a) Distributions of dataset features as ellipses with $1\times$ standard deviation

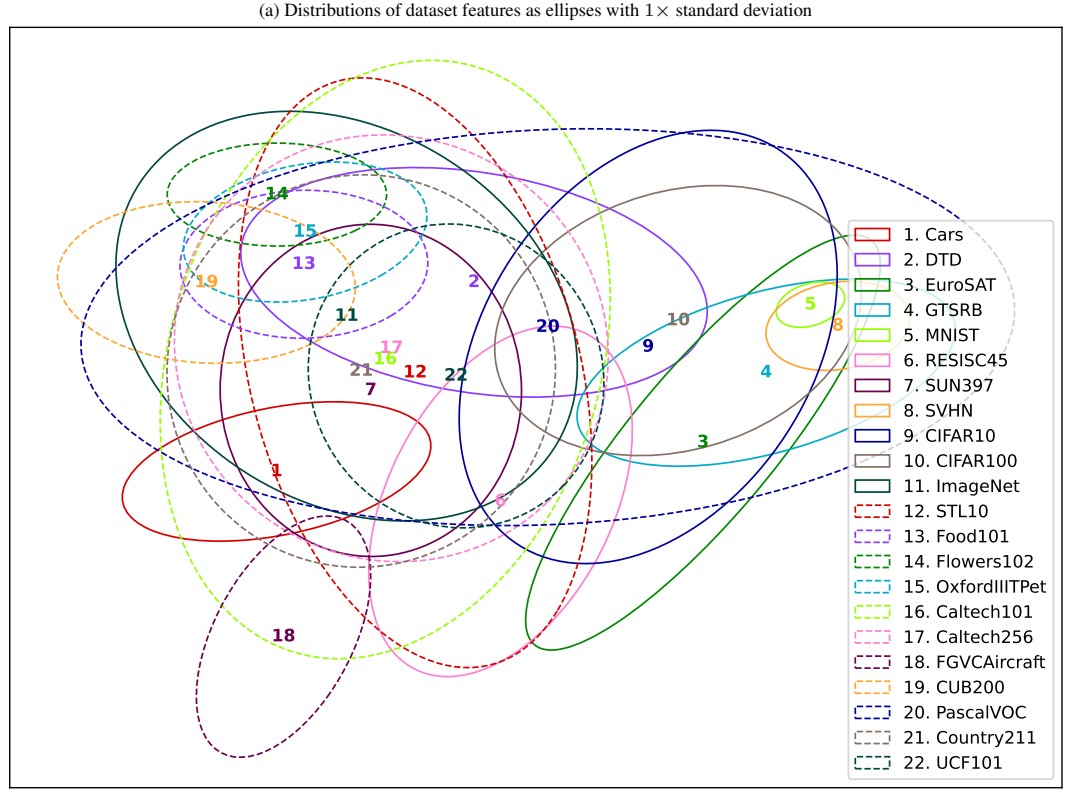

(b) Distributions of dataset features as ellipses with $3\times$ standard deviation

Figure 8: visualisation of dataset image feature distributions as ellipses. The mean image features for all datasets are visualised as the ellipse center, with the dimensionality reduced to 2 using Principal Component Analysis (PCA). The dimensionality of covariance matrices are also reduced using the same principal components. We show visualisations with (a) $\times 1$ and (b) $\times 3$ standard deviations. Pre-trained CLIP [47] with ViT-B/32 is used to extract image features.

Table 6: Learning rates and training epochs for task negation.

| | Cars | DTD | EuroSAT | GTSRB | MNIST | RESISC45 | SUN397 | SVHN |
|---|---|---|---|---|---|---|---|---|
| Learning rate | $5 \cdot 10^{-3}$ | $10^{-2}$ | $5 \cdot 10^{-3}$ | $5 \cdot 10^{-3}$ | $3 \cdot 10^{-3}$ | $2 \cdot 10^{-3}$ | $3 \cdot 10^{-3}$ | $5 \cdot 10^{-3}$ |
| Epochs | 20 | 20 | 3 | 5 | 7 | 10 | 10 | 2 |

Table 7: Accuracy on target and control tasks of task negation for each of the eight datasets. Highest performance in each section is highlighted in bold. The method *search* corresponds to model $f(x; \boldsymbol{\theta}_0 + \alpha \boldsymbol{\tau}_t)$, where $\alpha$ is determined via a hyper-parameter search. Our method *aniso.* corresponds to model $f(x; \boldsymbol{\theta}_0 + \Lambda_t \boldsymbol{\tau}_t)$, where $\Lambda_t$ is a learnable scaling matrix.

| | | Cars | | DTD | | EuroSAT | | GTSRB | | MNIST | | RESISC45 | | SUN397 | | SVHN | | Average | |
|---|---|---|---|---|---|---|---|---|---|---|---|---|---|---|---|---|---|---|---|
| | | Tgt. | Ctr. | Tgt. | Ctr. | Tgt. | Ctr. | Tgt. | Ctr. | Tgt. | Ctr. | Tgt. | Ctr. | Tgt. | Ctr. | Tgt. | Ctr. | Tgt. | Ctr. |
| ViT-B/32 | Zero-shot | 59.73 | 63.35 | 43.99 | 63.35 | 45.19 | 63.35 | 32.56 | 63.35 | 48.25 | 63.35 | 60.65 | 63.35 | 63.18 | 63.35 | 31.61 | 63.35 | 48.18 | 63.35 |
| | Std. (search) | 35.06 | **60.72** | 29.41 | **60.66** | 11.89 | 60.68 | 7.16 | 60.39 | 12.67 | 60.84 | 31.27 | **61.26** | 51.25 | 60.48 | **7.03** | 60.61 | 23.22 | 60.71 |
| | Std. (aniso.) | **28.95** | 60.52 | **25.21** | 60.48 | **10.44** | **61.62** | **5.51** | **60.67** | **10.76** | **62.9** | **20.95** | 60.72 | **46.29** | **60.82** | 7.28 | **62.72** | **18.76** | **61.21** |
| | Lin. (search) | 27.06 | **60.71** | 15.27 | 60.42 | **0.26** | 60.63 | **1.03** | **61.23** | 0.06 | **62.52** | 6.83 | **60.68** | 41.3 | 59.93 | **0.54** | 59.77 | 11.54 | 60.74 |
| | Lin. (aniso.) | **23.96** | 60.57 | **15.05** | **60.55** | 0.44 | **61.86** | 1.1 | 61.16 | **0.06** | 61.71 | **4.48** | 60.26 | 42.71 | **60.78** | 0.76 | **61.2** | **11.06** | **61.02** |
| ViT-B/16 | Zero-shot | 64.61 | 68.33 | 45.11 | 68.33 | 55.78 | 68.33 | 43.34 | 68.33 | 51.79 | 68.33 | 65.76 | 68.33 | 65.5 | 68.33 | 51.98 | 68.33 | 55.48 | 68.33 |
| | Std. (search) | 24.19 | **64.41** | 21.65 | 63.75 | **12.41** | 64.76 | **7.16** | 63.95 | 9.85 | 65.52 | 25.48 | 64.23 | 47.86 | 64.16 | **6.47** | 66.47 | 19.38 | 64.66 |
| | Std. (aniso.) | **16.63** | 63.95 | **20.69** | **64.6** | 15.93 | **67.65** | 8.21 | **66.37** | **9.51** | **68.29** | **21.29** | **65.17** | **45.43** | **65.02** | 6.84 | **67.93** | **17.34** | **65.84** |
| | Lin. (search) | 23.91 | **64.74** | 11.01 | **64.89** | **0.15** | 64.12 | 3.06 | **66.99** | 0.21 | 67.41 | 5.48 | 64.52 | 42.39 | 65.11 | **0.88** | 66.54 | 10.88 | 65.54 |
| | Lin. (aniso.) | **19.85** | 64.05 | **9.68** | 64.57 | 0.33 | **65.91** | **0.97** | 65.51 | **0.01** | 66.87 | 7.27 | **65.39** | 41.18 | **65.36** | 1.97 | **67.01** | **10.16** | **65.58** |
| ViT-L/14 | Zero-shot | 77.75 | 75.54 | 55.32 | 75.54 | 61.33 | 75.54 | 50.55 | 75.54 | 76.36 | 75.54 | 71.05 | 75.54 | 68.28 | 75.54 | 58.45 | 75.54 | 64.89 | 75.54 |
| | Std. (search) | 24.44 | **71.34** | 26.91 | 71.83 | **8.63** | 71.46 | 6.24 | **71.78** | 11.15 | 72.43 | **17.98** | 72.07 | 51.11 | 71.99 | **6.72** | 73.53 | 19.15 | 72.05 |
| | Std. (aniso.) | **14.49** | 71.07 | **23.94** | **72.2** | 12.15 | **74.81** | **3.95** | 71.66 | **7.29** | **74.69** | 25.11 | **74.29** | 47.93 | **72.8** | 7.16 | **74.69** | **17.75** | **73.28** |
| | Lin. (search) | 18.57 | 71.09 | 13.03 | **71.92** | **0.33** | 73.15 | 5.57 | **74.41** | 5.31 | 74.32 | **3.11** | 72.03 | **45.79** | 72.2 | **10.54** | 74.51 | 12.78 | 72.95 |
| | Lin. (aniso.) | **16.9** | **71.67** | **10.48** | 71.78 | 1.19 | **74.49** | **4.13** | 74.39 | 7.6 | **74.98** | 6.46 | **73.38** | 44.96 | **72.4** | 13.23 | **75.18** | **12.61** | **73.14** |

# B    Task negation

The evaluation of task negation is conducted on eight classification datasets (1–8 in Table 5), following previous practice [28, 44]. In particular, we learn anisotropic scaling using the validation set of each dataset. We also adjust the learning rates and training epochs on the same validation set. The details are shown in Table 6. We report detailed task negation results for each dataset in Table 7. In addition, for more evidence that weight matrices learn large negative coefficients, we show a detailed visualisation of the learned coefficients in Figure 9 and distribution of the coefficients in Figure 10.

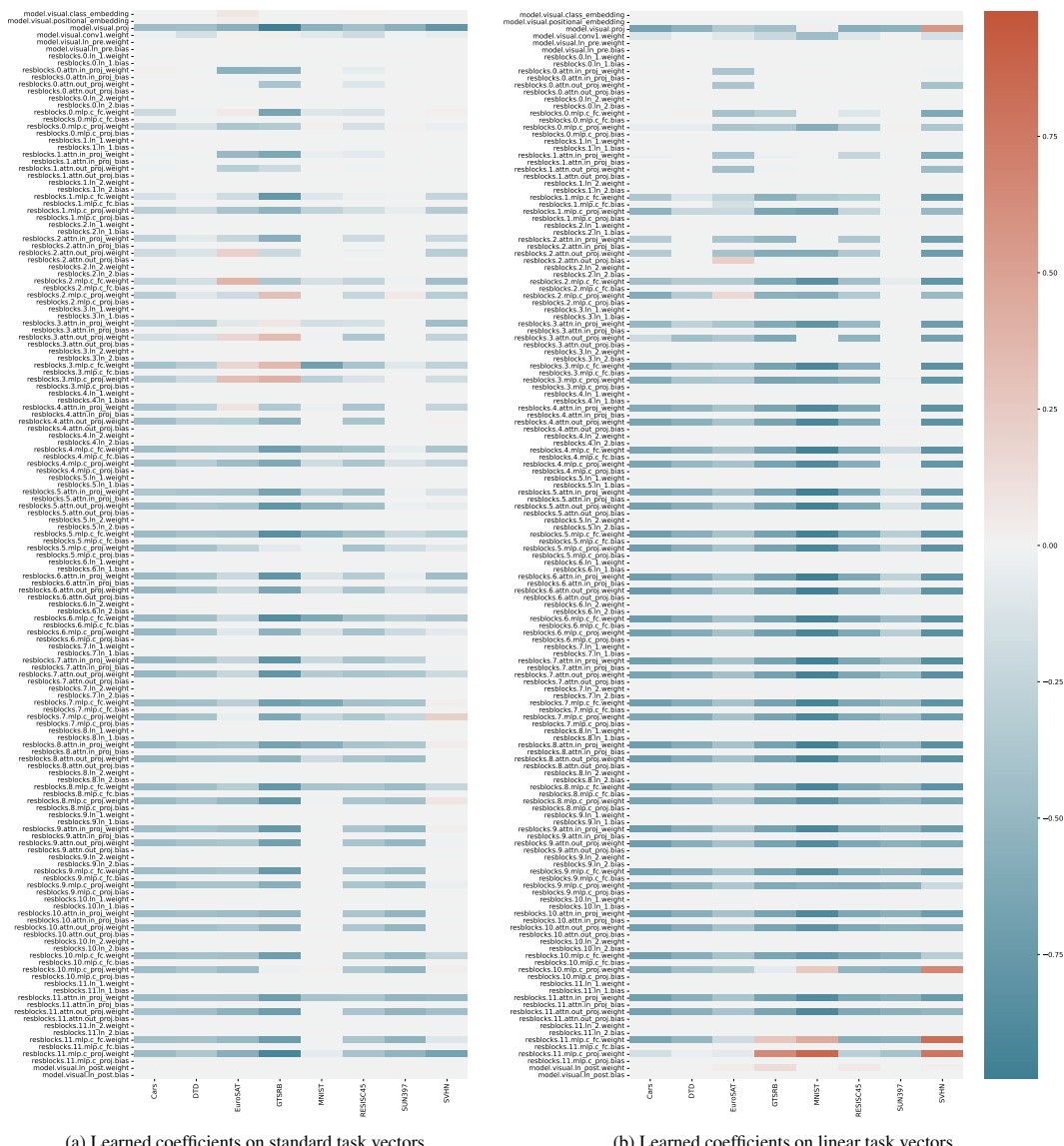

(a) Learned coefficients on standard task vectors      (b) Learned coefficients on linear task vectors

Figure 9: visualisation of the learned coefficients for (a) standard and (b) linear task vectors in task negation. Note that coefficients for different datasets are learned independently, despite being visualised jointly. Large negative coefficients can be observed on weight matrices. CLIP with ViT-B/32 backbone is used.

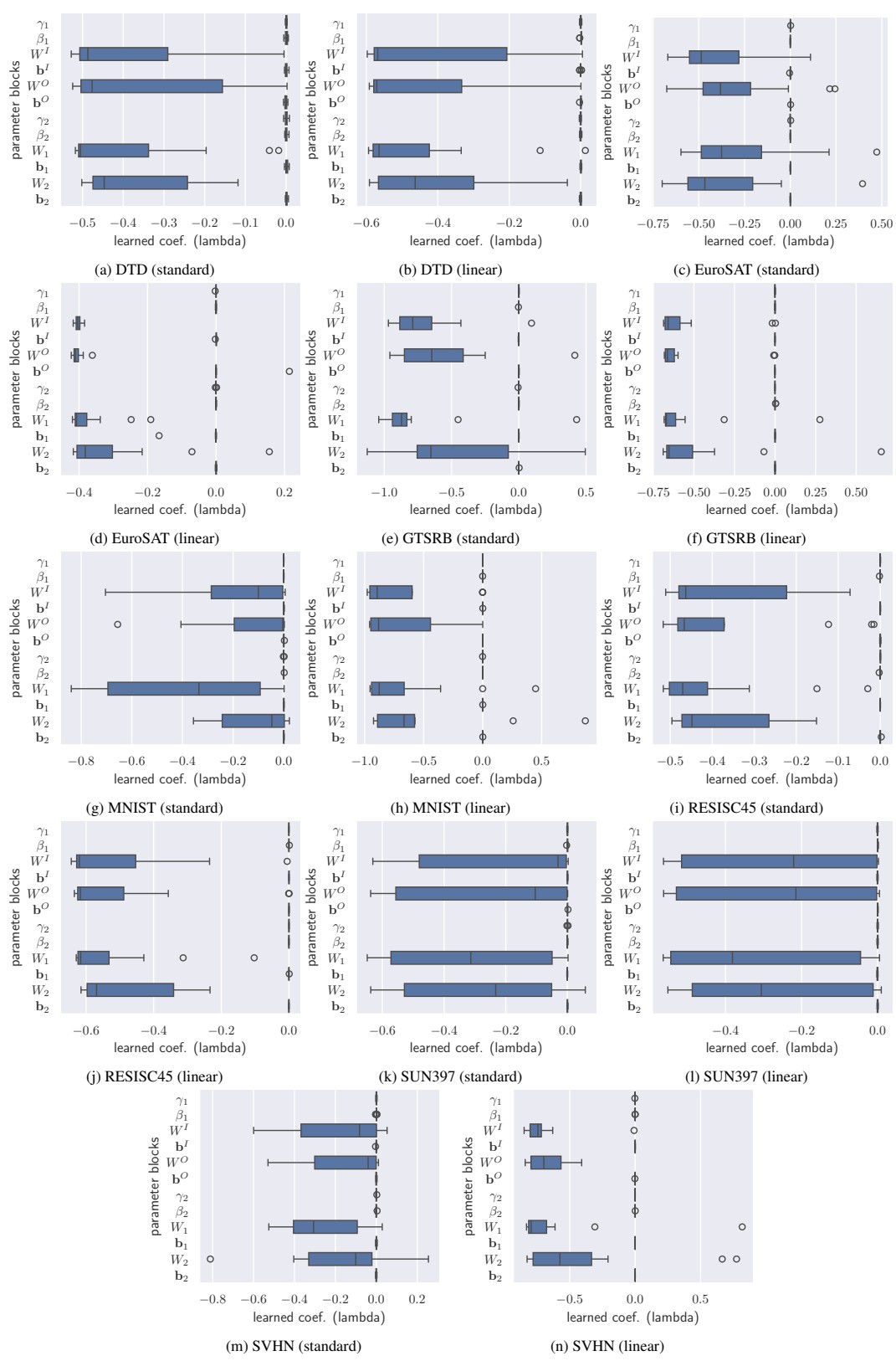

Figure 10: Additional box-and-whisker plots for the learned coefficients in task negation, beside previous visualisation on Cars (Figure 3a), including results on (a, b) DTD, (c, d) EuroSAT, (e, f) GTSRB, (g, h) MNIST, (i, j) RESISC45, (k, l) SUN397 and (m, n) SVHN.

Table 8: Detailed performance for task addition across all eight datasets. We show additional performance with learned isotropic scaling, which have comparable accuracy to the simple hyper-parameter search used in previous methods [28, 44]. Highest performance in each section is highlighted in bold. The method *search* corresponds to model $f(x; \theta_0 + \alpha \sum \tau_i)$, where $\alpha$ is determined via a hyper-parameter search. Methods *iso.* and *aniso.* use learned coefficients and correspond to models $f(x; \theta_0 + \sum \alpha_i \tau_i)$ and $f(x; \theta_0 + \sum \Lambda_i \tau_i)$, respectively.

| | | Cars | | DTD | | EuroSAT | | GTSRB | | MNIST | | RESISC45 | | SUN397 | | SVHN | | Average | |
|---|---|---|---|---|---|---|---|---|---|---|---|---|---|---|---|---|---|---|---|
| | | Abs. | Rel. | Abs. | Rel. | Abs. | Rel. | Abs. | Rel. | Abs. | Rel. | Abs. | Rel. | Abs. | Rel. | Abs. | Rel. | Abs. | Rel. |
| ViT-B/32 | Zero-shot | 59.73 | - | 43.99 | - | 45.19 | - | 32.56 | - | 48.25 | - | 60.65 | - | 63.18 | - | 31.61 | - | 48.14 | - |
| | Std. (search) | 58.97 | 75.35 | 52.29 | 66.24 | 80.04 | 80.94 | 66.74 | 67.31 | 95.96 | 96.3 | 70.54 | 73.53 | 60.74 | 80.55 | 75.66 | 77.69 | 70.12 | 77.24 |
| | Std. (iso.) | 56.88 | 72.68 | 51.97 | 65.84 | 87.96 | 88.95 | 73.14 | 73.77 | 82.23 | 82.52 | 61.16 | 63.75 | 62.88 | 83.4 | 87.98 | 90.34 | 70.53 | 77.66 |
| | Std. (aniso.) | **71.56** | **91.43** | **72.66** | **92.05** | **93.93** | **94.98** | **89.8** | **90.58** | **96.13** | **96.47** | **87.71** | **91.43** | **68.9** | **91.37** | **91.94** | **94.41** | **84.98** | **93.79** |
| | Lin. (search) | 67.14 | 87.97 | 58.56 | 73.2 | 95.67 | 97.62 | 67.58 | 72.13 | 94.61 | 95.25 | 81.25 | 85.92 | 63.99 | 83.78 | 68.36 | 85.5 | 74.67 | 85.17 |
| | Lin. (iso.) | 63.4 | 83.07 | 65.59 | 81.98 | 93.78 | 95.69 | 82.42 | 87.97 | 95.24 | 95.88 | 84.78 | 89.64 | 58.79 | 76.96 | 67.92 | 84.95 | 76.49 | 87.02 |
| | Lin. (aniso.) | **72.33** | **94.77** | **73.03** | **91.29** | **95.26** | **97.2** | **88.41** | **94.36** | **97.27** | **97.93** | **88.79** | **93.89** | **68.53** | **89.71** | **72.53** | **90.71** | **83.42** | **95.42** |
| ViT-B/16 | Zero-shot | 64.61 | - | 45.11 | - | 55.78 | - | 43.34 | - | 51.79 | - | 65.76 | - | 65.5 | - | 51.98 | - | 55.48 | - |
| | Std. (search) | 68.47 | 81.38 | 52.82 | 64.48 | 75.0 | 75.81 | 71.03 | 71.86 | 96.97 | 97.27 | 76.35 | 79.05 | 66.57 | 85.23 | 81.82 | 83.75 | 73.63 | 79.85 |
| | Std. (iso.) | 65.55 | 77.9 | 50.05 | 61.1 | 81.96 | 82.85 | 74.06 | 74.93 | 94.96 | 95.26 | 80.94 | 83.8 | 60.48 | 77.43 | 91.65 | 93.81 | 74.96 | 80.88 |
| | Std. (aniso.) | **71.79** | **85.32** | **67.07** | **81.88** | **91.85** | **92.85** | **91.9** | **92.98** | **97.02** | **97.32** | **84.3** | **87.28** | **66.6** | **85.26** | **92.66** | **94.85** | **86.08** | **93.44** |
| | Lin. (search) | 72.7 | 85.26 | 60.96 | 73.65 | 95.0 | 96.79 | 70.39 | 74.78 | 95.37 | 96.17 | 83.78 | 87.63 | 71.47 | 90.67 | 70.44 | 84.71 | 77.51 | 86.21 |
| | Lin. (iso.) | 72.6 | 85.14 | 70.48 | 85.15 | 92.19 | 93.92 | 82.43 | 87.58 | 93.86 | 94.65 | 88.14 | 92.2 | 61.26 | 77.72 | 72.95 | 87.73 | 79.24 | 88.01 |
| | Lin. (aniso.) | **81.78** | **95.9** | **77.13** | **93.19** | **96.33** | **98.15** | **88.65** | **94.18** | **98.6** | **99.43** | **90.49** | **94.65** | **72.95** | **92.55** | **77.11** | **92.72** | **85.38** | **95.10** |
| ViT-L/14 | Zero-shot | 77.75 | - | 55.32 | - | 61.33 | - | 50.55 | - | 76.36 | - | 71.05 | - | 68.28 | - | 58.45 | - | 64.89 | - |
| | Std. (search) | 81.69 | 89.12 | 64.63 | 76.27 | 88.67 | 88.83 | 93.88 | 94.54 | 98.75 | 98.98 | 82.68 | 85.11 | 71.3 | 86.98 | 81.86 | 83.56 | 82.93 | 87.92 |
| | Std. (iso.) | 85.72 | 93.52 | 71.38 | 84.24 | 83.74 | 83.9 | 91.74 | 92.39 | 96.5 | 96.72 | 91.56 | 94.25 | 60.33 | 73.59 | 94.94 | 96.91 | 84.49 | 89.44 |
| | Std. (aniso.) | **89.58** | **97.72** | **80.85** | **95.42** | **98.0** | **98.18** | **96.75** | **97.43** | **98.48** | **98.71** | **93.03** | **95.77** | **77.96** | **95.1** | **96.24** | **98.23** | **91.36** | **97.07** |
| | Lin. (search) | 85.13 | 94.86 | 74.41 | 89.28 | 95.89 | 97.33 | 77.82 | 81.12 | 98.11 | 98.75 | 89.87 | 93.14 | 74.29 | 90.0 | 82.45 | 90.38 | 84.75 | 91.86 |
| | Lin. (iso.) | 84.18 | 93.81 | 74.41 | 89.28 | 94.89 | 96.32 | 82.62 | 86.13 | 97.16 | 97.8 | 91.33 | 94.65 | 73.87 | 89.48 | 83.01 | 90.99 | 85.18 | 92.31 |
| | Lin. (aniso.) | **87.38** | **97.37** | **78.51** | **94.19** | **95.7** | **97.14** | **91.73** | **95.62** | **98.39** | **99.03** | **93.56** | **96.96** | **77.25** | **93.58** | **86.7** | **95.04** | **88.65** | **96.12** |

# C   Task addition

Task addition is also evaluated on datasets 1–8 shown in Table 5. The hyper-parameters are identical to fine-tuning, except the learning rate is modified to $10^{-3}$. We show detailed performance on each dataset in Table 8, where we compare our method against hyper-parameter search used in previous works [28, 44], and another variant with learned isotropic scaling. We also visualise the learned coefficients with $L_1$ regularisation in Figure 12. It can be easily observed that weight matrices, particularly those in the deeper layers, have significantly higher learned coefficients, which conforms to our observations in Figures 3b and 3c.

## C.1   Comparison against full-parameter optimisation

Since our method involves learning the coefficients, unlike previous methods [28, 44] that only require a hyper-parameter search, we also compare against the direct fine-tuning approach. We fine-tune the pre-trained model on the union of eight datasets, assuming only the validation sets are available. The results are shown in Figure 11. Unsurprisingly, task vector compositions, whether the coefficients are searched or learned, are less susceptible to the lack of data, as the accuracy only starts to drop with less than 35% of the data. The performance of full-parameter fine-tuning, however, drops substantially as the amount of data available decreases.

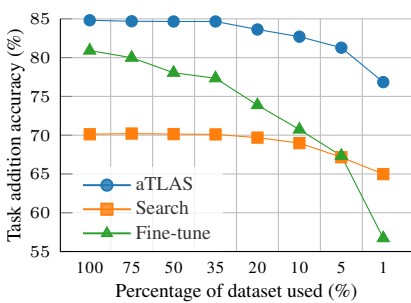

Figure 11: Task addition accuracy averaged across eight datasets (1–8) versus different percentage of validation data used. Standard task vectors are used.

## C.2   Disentanglement error

In addition, we provide more technical details and intuitions on the pairwise disentanglement error [44], which was visualised in Figure 4. Specifically, we make a few changes to the formulation proposed by Ortiz-Jiménez et al. [44], and evaluate the disentanglement error only with the optimal

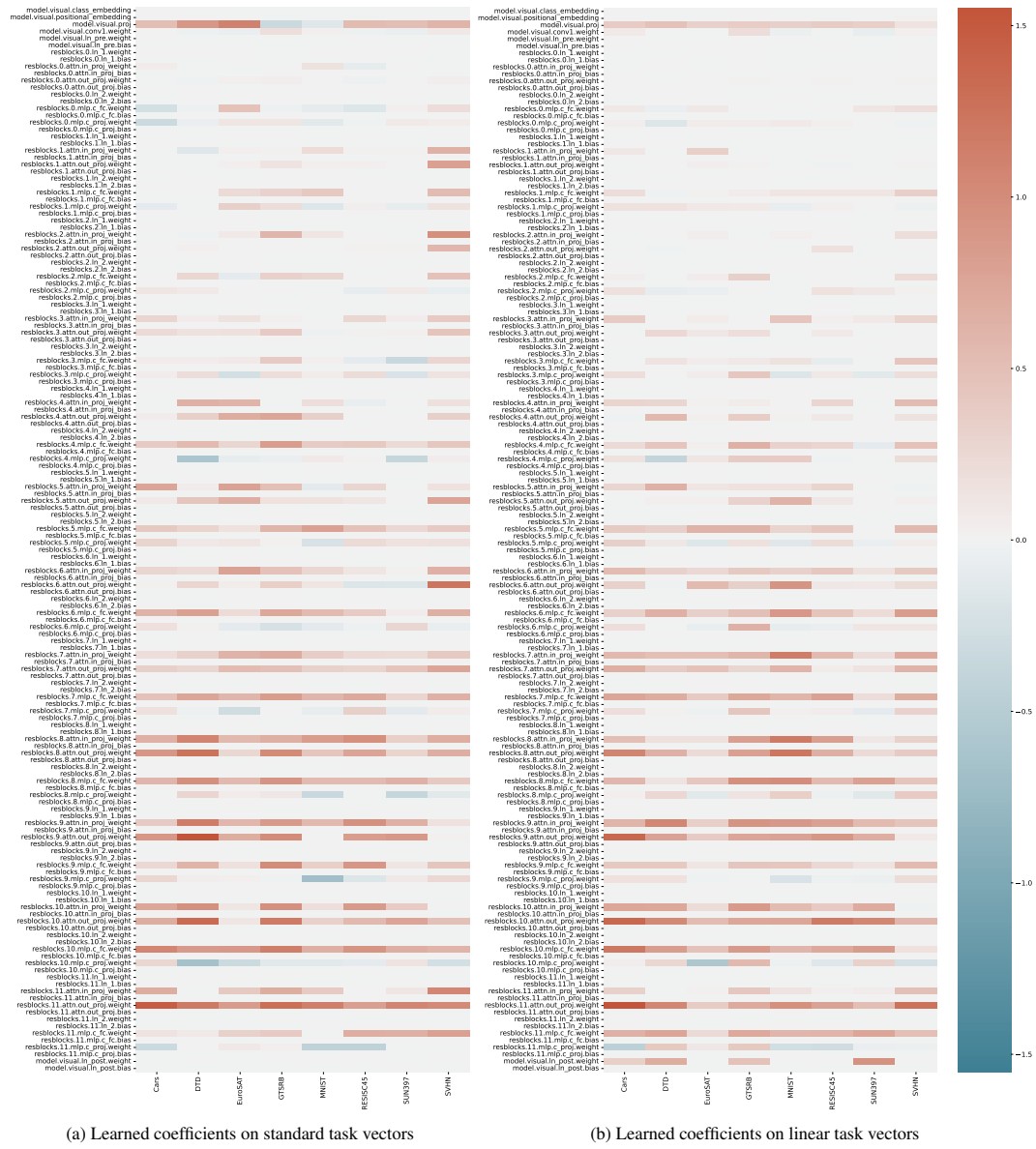

(a) Learned coefficients on standard task vectors      (b) Learned coefficients on linear task vectors

Figure 12: visualisation of the learned coefficients for (a) standard and (b) linear task vectors in task addition. Note that $L_1$ regularisation has been applied to the coefficients during training for better clarity. CLIP with ViT-B/32 backbone is used to produce the results.

coefficients. Given two datasets $\mathcal{D}_1, \mathcal{D}_2$ and the respective task vectors $\boldsymbol{\tau}_1, \boldsymbol{\tau}_2$, we overload the definition of function $f$ to denote the mapping from data space $\mathcal{X}$ to the label space $\mathcal{Y}$, and define the disentanglement error as

$$\xi(\boldsymbol{\tau}_1, \boldsymbol{\tau}_2) = \mathbf{E}_{\mathbf{x} \in \mathcal{D}_1}\big[\delta\big(f(\mathbf{x}; \boldsymbol{\theta}_0 + \Lambda_1^\star \boldsymbol{\tau}_1), f(\mathbf{x}; \boldsymbol{\theta}_0 + \Lambda_1^\star \boldsymbol{\tau}_1 + \Lambda_2^\star \boldsymbol{\tau}_2)\big)\big], \tag{10}$$

$$\xi(\boldsymbol{\tau}_2, \boldsymbol{\tau}_1) = \mathbf{E}_{\mathbf{x} \in \mathcal{D}_2}\big[\delta\big(f(\mathbf{x}; \boldsymbol{\theta}_0 + \Lambda_2^\star \boldsymbol{\tau}_2), f(\mathbf{x}; \boldsymbol{\theta}_0 + \Lambda_1^\star \boldsymbol{\tau}_1 + \Lambda_2^\star \boldsymbol{\tau}_2)\big)\big], \tag{11}$$

where $\Lambda_1^\star, \Lambda_2^\star$ are the learned coefficients in task addition, and $\delta$ is defined as

$$\delta(x_1, x_2) = \begin{cases} 0 & x_1 = x_2, \\ 1 & x_1 \neq x_2. \end{cases} \tag{12}$$

The error metric $\xi(\boldsymbol{\tau}_1, \boldsymbol{\tau}_2)$ measures the percentage of data in dataset $\mathcal{D}_1$, such that when a second task vector $\boldsymbol{\tau}_2$ is added to the model, the predicted labels differ from when only using task vector $\boldsymbol{\tau}_1$. As task vector $\boldsymbol{\tau}_1$ is acquired from dataset $\mathcal{D}_1$, a low disentanglement error indicates that most predictions made by $\boldsymbol{\tau}_1$—highly likely to be correct—will be retained, thus resulting in higher performance in task addition.

Table 9: Average accuracy for few-shot recognition over 22 datasets. We report accuracy averaged over 3 random n-shot sample selections, with $1\times$ standard error. Results are produced using CLIP with ViT-B/32 backbone. For our method, we show results with both standard [28] and linearised [44] task vectors. The best method for each choice of $k \in \{1, 2, 4, 8, 16\}$ is highlighted in bold.

| Shots ($k$) | Tip-Adapter | LP++ | aTLAS | | aTLAS w/ LP++ | | aTLAS w/ Tip-Adapter | |
|---|---|---|---|---|---|---|---|---|
| | | | Std. | Lin. | Std. | Lin. | Std. | Lin. |
| 1 | $64.3 \pm 0.2$ | $64.5 \pm 0.1$ | $66.2 \pm 0.2$ | $64.6 \pm 0.1$ | $68.9 \pm 0.2$ | $\mathbf{69.3} \pm 0.1$ | $68.7 \pm 0.4$ | $66.7 \pm 0.3$ |
| 2 | $67.0 \pm 0.1$ | $67.3 \pm 0.1$ | $67.6 \pm 0.1$ | $65.4 \pm 0.1$ | $71.5 \pm 0.1$ | $67.2 \pm 0.2$ | $\mathbf{71.9} \pm 0.2$ | $68.9 \pm 0.1$ |
| 4 | $69.7 \pm 0.1$ | $69.9 \pm 0.1$ | $70.0 \pm 0.0$ | $66.6 \pm 0.1$ | $73.7 \pm 0.1$ | $70.8 \pm 0.1$ | $\mathbf{74.3} \pm 0.1$ | $71.6 \pm 0.2$ |
| 8 | $71.8 \pm 0.1$ | $72.3 \pm 0.1$ | $71.5 \pm 0.0$ | $68.2 \pm 0.1$ | $75.8 \pm 0.1$ | $73.5 \pm 0.1$ | $\mathbf{76.5} \pm 0.1$ | $74.2 \pm 0.1$ |
| 16 | $73.7 \pm 0.1$ | $74.1 \pm 0.1$ | $72.9 \pm 0.1$ | $69.8 \pm 0.1$ | $77.8 \pm 0.0$ | $76.2 \pm 0.1$ | $\mathbf{78.0} \pm 0.1$ | $76.7 \pm 0.0$ |

# D  Few-shot learning

## D.1  Baselines: Tip-Adapter and LP++

Two variants of Tip-Adapter [69] were proposed for few-shot recognition where the weights of the adaptor are either fixed based on features of the few-shot examples or further fine-tuned. We only study the fine-tuned variant due to its higher performance. Tip-Adapter has two hyper-parameters, which in the original paper are optimised through hyper-parameter search on a separate validation set. This practice may not align with the principles of few-shot learning, where access to extensive validation data is typically limited. In addition, Huang et al. [25] note that the performance of Tip-Adapter is very sensitive to these hyper-parameters. We thus opt to learn these two hyper-parameters together with the feature adaptor through gradient descent. The learning rates for the feature adaptor and the hyper-parameters are set to $10^{-3}$ and $10^{-1}$, respectively.

For both Tip-Adapter and LP++ [25], we conduct experiments using the publicly available codebase [5]. We train both LP++ and Tip-Adapter for 300 epochs on frozen zero-shot features. We apply a cosine annealing decay for Tip-Adapter and maintain fixed learning rates for LP++ as per the official implementation.

## D.2  linearised task vectors

We report the average few-shot accuracy over the 22 datasets in Table 9, which corresponds to results in Figure 5a. In particular, we show results with linearised task vectors, as proposed by Ortiz-Jiménez et al. [44]. As highlighted in Section 4, learned anisotropic scaling allows standard task vectors to achieve stronger performance than the linear variants in task addition. For few-shot recognition, we again observe that standard task vectors result in superior performance in most cases. We, however, note the exception that linear task vectors when combined with LP++ achieve higher performance in the 1-shot setting. Nevertheless, the margin over standard task vectors is not very significant, and aTLAS using standard task vectors when integrated with Tip-Adapter is generally a stronger few-shot model.

## D.3  Integrating state-of-the-art methods into aTLAS

We use the AdamW [38] optimiser with a learning rate of $10^{-1}$ and a weight decay of $10^{-1}$. Our method by itself is trained for 10 epochs with ViT backbones and 30 epochs with ResNet backbones.

We show that state-of-the-art few-shot methods can be seamlessly integrated into our method, since both Tip-Adapter and LP++ focus on the classifier, while aTLAS improves the feature representations. We experiment with two strategies to combine aTLAS with previous methods, where we either (1) train our method first and use the frozen representations to train a previous method, or (2) train parameters in both methods jointly. Results in Table 10 shows that the joint training strategy results in higher performance, particularly in low-shot settings. We therefore adopt the joint training strategy when combing our method with Tip-Adapter. During training, we adopt different learning rates for different parameter groups, that is, $10^{-1}$ for learnable coefficients in aTLAS and the hyper-parameters in Tip-Adapter, and $10^{-3}$ for the adaptor. The joint training takes 20 epochs for ViT backbones and 60 epochs on ResNet backbones, twice the number of epochs when training aTLAS alone.

---

[5] github.com/fereshteshakeri/fewshot-clip-strong-baseline

Table 10: Comparison of few-shot recognition accuracy between training our method and Tip-Adapter sequentially and jointly over different shots ($k$). ViT-B/32 is used as the backbone. Results are averaged across three random seeds. Highest performance in each section is highlighted in bold.

| $k$ | Strategy | Cars | DTD | EuroSAT | GTSRB | MNIST | RESISC45 | SUN397 | SVHN | CIFAR10 | CIFAR100 | ImageNet | STL10 | Food101 | Caltech256 | FGVCAircraft | Flowers102 | OxfordIIITPet | CUB200 | PascalVOC | Country211 | Caltech101 | UCF101 | Average |
|---|---|---|---|---|---|---|---|---|---|---|---|---|---|---|---|---|---|---|---|---|---|---|---|---|
| 1 | Seq. | 59.7 | 43.7 | 64.5 | 44.6 | 75.7 | 63.6 | 64.0 | 38.3 | 89.8 | 69.2 | 63.7 | 97.2 | 84.0 | **84.9** | 19.9 | 70.2 | **88.8** | 53.0 | 76.2 | 17.2 | **89.5** | 66.0 | 64.7 |
|   | Joint | **61.4** | **49.3** | **72.5** | **57.5** | **84.8** | **70.8** | **67.5** | **57.0** | **91.0** | **75.5** | **64.2** | 97.2 | 84.0 | 84.0 | **22.6** | **79.0** | 88.7 | **54.4** | 76.2 | **17.3** | 89.3 | **69.0** | **68.8** |
| 2 | Seq. | 63.6 | 48.8 | 72.6 | 49.0 | 85.4 | 72.8 | 67.8 | 42.4 | 89.8 | 69.9 | **64.8** | 97.2 | 84.0 | 86.1 | 22.8 | 78.4 | 88.8 | 57.5 | 76.2 | 17.2 | 90.3 | 69.8 | 68.0 |
|   | Joint | **65.1** | **55.9** | **83.4** | **67.6** | **86.7** | **77.3** | **69.2** | **56.3** | **93.4** | **75.9** | 64.5 | **97.5** | 84.0 | **88.0** | **26.2** | **85.0** | **89.6** | **58.8** | 76.2 | **18.1** | **90.4** | **72.9** | **71.9** |
| 4 | Seq. | 68.7 | 55.5 | 76.8 | 58.3 | 88.4 | 75.2 | 70.3 | 45.5 | 90.8 | 71.9 | **65.8** | **98.0** | 84.0 | 86.6 | 28.9 | 86.4 | 89.0 | **63.6** | 76.2 | 17.2 | **92.2** | 73.2 | 71.0 |
|   | Joint | **69.0** | **63.7** | **79.6** | **73.1** | **90.6** | **78.9** | **70.8** | **67.6** | **93.9** | **78.3** | 64.7 | 97.5 | 84.0 | **87.5** | **29.3** | **91.6** | **89.7** | 62.8 | **77.3** | **18.1** | 90.4 | **77.5** | **74.4** |
| 8 | Seq. | 73.7 | 64.7 | 82.3 | 69.5 | 91.0 | 79.8 | **72.3** | 43.0 | 91.9 | 73.4 | **66.9** | 97.7 | 84.0 | **88.9** | 34.7 | 91.2 | **90.8** | **69.3** | 76.2 | 18.5 | **91.9** | 78.8 | 74.1 |
|   | Joint | **74.2** | **69.4** | **89.5** | **78.4** | **94.0** | **81.1** | 72.1 | **60.1** | **95.0** | **78.5** | 65.6 | **97.8** | **84.3** | 88.6 | **36.6** | **93.7** | 90.0 | 69.0 | **77.6** | **19.0** | 91.7 | **78.9** | **76.6** |
| 16 | Seq. | 77.6 | **69.0** | 89.3 | 79.3 | 92.8 | **83.7** | **74.1** | **64.6** | 94.0 | 79.9 | 66.8 | 97.5 | **84.6** | **89.2** | 36.1 | 94.2 | **91.4** | 72.8 | **80.3** | 20.1 | **93.6** | 80.0 | 77.8 |
|   | Joint | **79.1** | 68.8 | **91.4** | **82.7** | **93.9** | 82.8 | 73.7 | 63.7 | **95.1** | **80.0** | **67.0** | **98.1** | 84.3 | 88.6 | **39.1** | **94.5** | 90.1 | **74.0** | 77.0 | **20.4** | 91.9 | **80.4** | **78.0** |

Table 11: Detailed few-shot accuracy for each dataset over different shots ($k$) using ViT-B/32 backbone. We report results averaged over 3 random seeds. In the case where the results are worse than the zero-shot accuracy, we report zero-shot accuracy. Highest performance and those within a range of 0.1 in each section are highlighted in bold. Tip-Adapter is abbreviated as Tip.

| $k$ | Method | Cars | DTD | EuroSAT | GTSRB | MNIST | RESISC45 | SUN397 | SVHN | CIFAR10 | CIFAR100 | ImageNet | STL10 | Food101 | Caltech256 | FGVCAircraft | Flowers102 | OxfordIIITPet | CUB200 | PascalVOC | Country211 | Caltech101 | UCF101 | Average |
|---|---|---|---|---|---|---|---|---|---|---|---|---|---|---|---|---|---|---|---|---|---|---|---|---|
| 0 | CLIP | 59.7 | 43.7 | 42.9 | 32.7 | 48.3 | 60.6 | 63.2 | 31.3 | 89.8 | 64.2 | 63.4 | 97.2 | 84.0 | 82.0 | 19.6 | 66.6 | 87.5 | 53.0 | 76.2 | 17.2 | 84.0 | 61.6 | 60.4 |
| 1 | Tip | **62.6** | **52.3** | 55.2 | 37.5 | 61.3 | 67.2 | 66.5 | 34.7 | 89.8 | 66.4 | 63.9 | **97.7** | 84.0 | 84.8 | 22.0 | 80.6 | 87.6 | 55.1 | **76.7** | **17.3** | 87.5 | 68.2 | 64.5 |
|   | LP++ | 61.5 | 51.3 | 60.0 | 39.5 | 50.5 | 68.8 | 65.8 | 31.8 | 89.8 | 66.3 | 63.9 | 97.2 | **84.1** | 84.7 | 23.7 | **81.9** | 87.5 | 55.1 | 76.2 | 17.2 | 88.3 | **69.7** | 64.3 |
|   | aTLAS | 59.7 | 43.7 | **74.7** | 52.4 | 79.5 | 64.5 | 63.2 | 59.0 | 89.8 | 70.2 | 63.4 | 97.2 | 84.0 | 83.4 | 19.6 | 66.6 | 87.5 | 53.0 | 76.2 | 17.2 | 89.1 | 62.5 | 66.2 |
|   | aTLAS w/ LP++ | 62.2 | 50.2 | 72.5 | 52.8 | 84.0 | 69.9 | 67.2 | **64.6** | **92.1** | **75.6** | **64.4** | 97.2 | 84.0 | **85.4** | **24.1** | 81.6 | 87.5 | **56.2** | 76.2 | 17.2 | **89.5** | 69.4 | **69.2** |
|   | aTLAS w/ Tip | 61.4 | 49.3 | 72.5 | **57.5** | **84.8** | **70.8** | **67.5** | 57.0 | 91.0 | 75.5 | 64.2 | 97.2 | 84.0 | 84.0 | 22.6 | 79.0 | **88.7** | 54.4 | 76.2 | **17.3** | 89.3 | 69.0 | 68.8 |
| 2 | Tip | 64.1 | 57.4 | 70.8 | 43.7 | 66.0 | 73.1 | 68.3 | 31.8 | 90.0 | 66.6 | 64.5 | **97.9** | **84.3** | 85.6 | 24.0 | **87.3** | 88.0 | 57.4 | 76.2 | 17.5 | 88.4 | 72.5 | 67.1 |
|   | LP++ | 64.2 | 57.3 | 69.8 | 42.3 | 69.7 | 74.7 | 67.4 | 31.3 | 89.8 | 67.6 | 64.5 | 97.4 | 84.0 | 86.3 | 25.6 | 86.8 | 88.7 | 59.7 | 76.2 | 17.4 | **90.4** | 72.3 | 67.4 |
|   | aTLAS | 59.7 | 43.9 | 79.5 | 53.8 | 86.0 | 68.0 | 64.0 | **60.9** | 91.1 | 72.5 | 63.9 | 97.2 | 84.0 | 84.6 | 21.2 | 67.4 | 88.4 | 53.0 | 76.2 | 17.2 | 89.8 | 65.0 | 67.6 |
|   | aTLAS w/ LP++ | **65.8** | **58.2** | 80.0 | 58.4 | **87.4** | 74.4 | 68.5 | 55.1 | 93.1 | **76.8** | **64.9** | 97.4 | 84.0 | 87.0 | **26.9** | 87.0 | 88.5 | **60.5** | 76.2 | 17.5 | 89.7 | 72.7 | 71.4 |
|   | aTLAS w/ Tip | 65.1 | 55.9 | **83.4** | **67.6** | 86.7 | **77.3** | **69.2** | 56.3 | **93.4** | 75.9 | 64.5 | 97.5 | 84.0 | **88.0** | 26.2 | 85.0 | **89.6** | 58.8 | 76.2 | **18.1** | **90.4** | **72.9** | **71.9** |
| 4 | Tip | 65.8 | 62.3 | 77.3 | 48.7 | 77.9 | 77.1 | 70.2 | 32.8 | 90.4 | 68.2 | 65.0 | 97.2 | **84.7** | 87.1 | 28.8 | **91.5** | 88.4 | 62.3 | 76.2 | 18.3 | 90.7 | 73.9 | 69.8 |
|   | LP++ | 67.5 | 61.5 | 74.2 | 54.0 | 72.5 | **79.1** | 70.5 | 34.3 | 90.8 | 68.1 | 65.4 | **98.0** | 84.3 | **88.7** | 27.4 | 90.1 | 87.7 | 62.5 | **77.3** | **18.8** | **91.8** | 75.5 | 70.0 |
|   | aTLAS | 60.9 | 48.6 | 84.3 | 66.3 | 89.6 | 72.0 | 65.1 | **71.8** | 91.9 | 75.1 | 64.7 | 97.2 | 84.0 | 85.2 | 22.9 | 69.9 | 88.2 | 53.5 | 76.2 | 17.2 | 90.7 | 65.2 | 70.0 |
|   | aTLAS w/ LP++ | **69.9** | 62.9 | **85.3** | **73.6** | 89.2 | 76.0 | **70.8** | 58.5 | 93.6 | 78.0 | **65.7** | 97.2 | 84.4 | 88.2 | 28.9 | 89.2 | **90.6** | **64.3** | 77.0 | 17.7 | 91.2 | 75.8 | 74.0 |
|   | aTLAS w/ Tip | 69.0 | **63.7** | 79.6 | 73.1 | **90.6** | 78.9 | **70.8** | 67.6 | **93.9** | **78.3** | 64.7 | 97.5 | 84.0 | 87.5 | **29.3** | **91.6** | 89.7 | 62.8 | **77.3** | 18.1 | 90.4 | **77.5** | **74.4** |
| 8 | Tip | 71.1 | 65.6 | 78.3 | 58.1 | 84.9 | 80.9 | 71.7 | 31.3 | 91.0 | 68.4 | 65.0 | 97.6 | **85.0** | 88.0 | 31.2 | 93.1 | **90.4** | 66.7 | 76.5 | **19.4** | 91.5 | 76.5 | 71.9 |
|   | LP++ | 72.2 | 65.2 | 79.4 | 61.2 | 82.5 | **81.8** | 72.2 | 31.3 | 91.0 | 69.6 | 66.9 | **97.9** | 84.7 | **89.1** | 31.2 | 92.2 | 89.6 | 66.7 | **78.7** | **19.3** | **92.9** | 76.8 | 72.4 |
|   | aTLAS | 61.6 | 52.1 | **90.8** | 67.2 | 90.2 | 74.6 | 65.9 | **72.2** | 93.0 | 77.3 | 64.8 | 97.2 | 84.0 | 85.8 | 24.8 | 73.0 | 89.9 | 55.4 | 77.0 | 17.4 | 91.2 | 67.8 | 71.5 |
|   | aTLAS w/ LP++ | 73.5 | 65.2 | 86.6 | 73.2 | 92.8 | 80.8 | **72.8** | 64.2 | 94.1 | **79.6** | **67.3** | 97.2 | 84.7 | 88.0 | 32.1 | 91.4 | 90.2 | 66.7 | 76.2 | 19.0 | 92.4 | 77.3 | 75.7 |
|   | aTLAS w/ Tip | **74.2** | **69.4** | 89.5 | **78.4** | **94.0** | 81.1 | 72.1 | 60.1 | **95.0** | 78.5 | 65.6 | 97.8 | 84.3 | 88.6 | **36.6** | **93.7** | 90.0 | **69.0** | 77.6 | 19.0 | 91.7 | **78.9** | **76.6** |
| 16 | Tip | 73.5 | 67.5 | 85.8 | 66.6 | 89.1 | 82.3 | 73.1 | 31.3 | 91.3 | 69.4 | 65.9 | 97.9 | 84.7 | 88.8 | 36.2 | **94.5** | 89.6 | 70.3 | 76.2 | **20.3** | 92.3 | 79.5 | 73.9 |
|   | LP++ | 75.2 | **69.7** | 86.4 | 64.8 | 87.4 | 83.3 | **74.4** | 31.3 | 91.7 | 70.8 | 68.2 | 98.0 | **85.5** | 89.1 | 35.7 | 92.2 | 90.6 | 69.3 | 76.9 | **20.4** | 93.1 | 79.5 | 74.2 |
|   | aTLAS | 62.9 | 55.5 | **92.6** | 71.3 | 92.8 | 77.0 | 66.5 | **78.5** | 93.9 | 77.8 | 65.4 | 97.3 | 84.3 | 86.6 | 24.9 | 74.1 | 90.6 | 56.5 | 76.9 | 17.8 | 92.8 | 68.4 | 72.9 |
|   | aTLAS w/ LP++ | 76.9 | 68.9 | 89.6 | 79.3 | **94.4** | **84.1** | 73.5 | 70.7 | 94.7 | **80.6** | **68.7** | 97.9 | 85.1 | **90.3** | 34.2 | 92.0 | **90.8** | 69.5 | **78.4** | 19.8 | **93.4** | 79.9 | 77.9 |
|   | aTLAS w/ Tip | **79.1** | 68.8 | 91.4 | **82.7** | 93.9 | 82.8 | 73.7 | 63.7 | **95.1** | 80.0 | 67.0 | **98.1** | 84.3 | 88.6 | **39.1** | **94.5** | 90.1 | **74.0** | 77.0 | **20.4** | 91.9 | **80.4** | **78.0** |

On the other hand, The joint training strategy with LP++ is non-trivial, due to LP++'s super-convergence strategy being designed around frozen feature representations, which would have been updated every iteration by aTLAS. We thus use the sequential strategy to combine aTLAS and LP++. We include detailed results for each dataset with ViT-B/32 in Table 11 and additional results with different backbones in Table 12, where we show our method scales well across different datasets and backbones.

Table 12: Detailed few-shot accuracy for each dataset across RN50, RN101, ViT-B/16, and ViT-L/14 backbones. We report results for the same random seed. In the case where the results are worse than the zero-shot accuracy, we report zero-shot accuracy. Highest performance and those within a range of 0.1 in each section are highlighted in bold. Tip-Adapter is abbreviated as Tip.

| Backbone | k | Method | Cars | DTD | EuroSAT | GTSRB | MNIST | RESISC45 | SUN397 | SVHN | CIFAR10 | CIFAR100 | ImageNet | STL10 | Food101 | Caltech256 | FGVCAircraft | Flowers102 | OxfordIIITPet | CUB200 | PascalVOC | Country211 | Caltech101 | UCF101 | Average |
|---|---|---|---|---|---|---|---|---|---|---|---|---|---|---|---|---|---|---|---|---|---|---|---|---|---|
| RN50 | 0 | CLIP | 54.3 | 41.2 | 41.5 | 35.1 | 58.1 | 53.1 | 60.1 | 28.9 | 71.5 | 40.3 | 59.9 | 94.2 | 80.6 | 77.3 | 17.0 | 66.2 | 85.8 | 46.5 | 66.2 | 15.4 | 77.6 | 58.3 | 55.9 |
| | 1 | Tip | 56.3 | 50.2 | 58.4 | 37.6 | 66.2 | 58.7 | 62.9 | 29.7 | 74.2 | 47.2 | 60.1 | 95.2 | 80.7 | 79.9 | 19.4 | 79.8 | 86.4 | 51.3 | 66.2 | 16.2 | 83.3 | 63.4 | 60.2 |
| | | LP++ | 57.2 | 44.8 | 58.2 | 35.9 | 57.4 | 59.4 | 63.0 | 30.9 | 72.2 | 45.0 | 60.7 | 94.2 | 80.6 | 80.9 | 21.2 | 80.9 | 86.0 | 53.8 | 68.0 | 16.2 | 85.4 | 64.7 | 59.9 |
| | | aTLAS | 54.3 | 41.2 | 62.7 | 44.5 | 75.6 | 56.0 | 61.0 | 50.7 | 78.4 | 54.0 | 60.7 | 95.2 | 80.6 | 79.8 | 19.5 | 66.2 | 85.8 | 50.5 | 66.2 | 15.4 | 86.8 | 60.0 | 61.1 |
| | | aTLAS w/ Tip | 56.5 | 44.2 | 44.9 | 37.5 | 78.3 | 54.9 | 62.0 | 29.0 | 71.6 | 40.8 | 63.5 | 97.2 | 84.0 | 85.2 | 23.5 | 75.7 | 87.4 | 53.5 | 76.4 | 17.4 | 88.7 | 68.4 | 60.9 |
| | 2 | Tip | 58.3 | 56.4 | 71.6 | 37.5 | 57.4 | 65.6 | 64.5 | 29.7 | 77.0 | 48.0 | 60.8 | 95.0 | 80.7 | 81.2 | 23.2 | 84.3 | 87.2 | 54.5 | 67.3 | 16.2 | 86.3 | 66.7 | 62.3 |
| | | LP++ | 60.0 | 52.7 | 66.1 | 41.7 | 61.5 | 64.6 | 64.8 | 29.7 | 72.1 | 46.9 | 61.5 | 94.4 | 80.6 | 82.5 | 22.6 | 84.2 | 86.6 | 56.0 | 67.5 | 16.2 | 88.1 | 66.1 | 62.1 |
| | | aTLAS | 55.2 | 42.0 | 76.6 | 52.3 | 73.0 | 59.8 | 60.6 | 56.8 | 79.7 | 57.0 | 60.7 | 94.3 | 80.6 | 80.4 | 20.6 | 67.0 | 85.8 | 51.0 | 66.2 | 15.8 | 86.9 | 59.6 | 62.8 |
| | | aTLAS w/ Tip | 58.9 | 53.1 | 52.5 | 63.3 | 88.8 | 62.6 | 64.3 | 28.9 | 72.1 | 41.0 | 63.7 | 97.1 | 84.0 | 86.5 | 24.9 | 85.9 | 87.5 | 57.2 | 76.5 | 17.5 | 89.0 | 73.0 | 64.9 |
| | 4 | Tip | 63.3 | 61.2 | 74.4 | 41.4 | 68.8 | 71.0 | 66.3 | 29.7 | 76.9 | 49.1 | 61.3 | 94.4 | 81.4 | 82.5 | 24.2 | 90.2 | 86.9 | 58.0 | 66.2 | 16.5 | 86.3 | 71.5 | 64.6 |
| | | LP++ | 63.2 | 59.5 | 76.6 | 43.4 | 70.4 | 69.8 | 66.8 | 29.7 | 74.9 | 48.0 | 62.4 | 94.2 | 81.1 | 83.7 | 24.6 | 89.2 | 86.9 | 60.1 | 66.7 | 16.2 | 87.0 | 70.9 | 64.8 |
| | | aTLAS | 57.6 | 43.0 | 79.7 | 55.3 | 82.2 | 62.0 | 61.4 | 53.4 | 81.6 | 58.9 | 61.4 | 94.2 | 80.6 | 80.9 | 22.6 | 69.7 | 86.5 | 52.3 | 66.8 | 15.4 | 87.1 | 61.3 | 64.3 |
| | | aTLAS w/ Tip | 62.9 | 58.9 | 64.2 | 72.0 | 90.1 | 70.0 | 65.1 | 29.8 | 71.7 | 42.1 | 64.1 | 97.4 | 84.2 | 87.0 | 29.2 | 91.2 | 88.7 | 63.1 | 76.2 | 17.5 | 90.1 | 76.0 | 67.8 |
| | 8 | Tip | 66.9 | 63.5 | 80.8 | 54.1 | 77.1 | 71.8 | 68.0 | 29.7 | 76.7 | 48.8 | 61.1 | 95.1 | 81.6 | 83.9 | 30.4 | 93.3 | 85.8 | 62.7 | 69.4 | 17.4 | 88.6 | 74.0 | 67.3 |
| | | LP++ | 68.2 | 64.0 | 78.7 | 50.6 | 76.7 | 74.7 | 70.0 | 29.7 | 75.5 | 50.3 | 63.8 | 95.8 | 81.6 | 84.3 | 28.8 | 92.8 | 88.0 | 64.2 | 68.5 | 17.0 | 88.7 | 74.1 | 67.5 |
| | | aTLAS | 58.4 | 48.5 | 80.0 | 57.1 | 85.4 | 66.9 | 62.3 | 60.8 | 84.2 | 61.2 | 61.9 | 95.8 | 81.1 | 82.2 | 23.6 | 71.4 | 87.8 | 53.0 | 69.1 | 15.6 | 89.6 | 63.5 | 66.3 |
| | | aTLAS w/ Tip | 69.9 | 61.1 | 74.4 | 81.1 | 91.5 | 75.9 | 65.2 | 28.9 | 73.6 | 45.5 | 64.7 | 97.1 | 84.0 | 87.8 | 34.0 | 93.3 | 89.6 | 68.0 | 76.2 | 17.6 | 90.7 | 78.9 | 70.4 |
| | 16 | Tip | 71.9 | 67.1 | 83.2 | 63.3 | 86.0 | 75.1 | 69.8 | 32.4 | 79.3 | 51.2 | 62.8 | 95.0 | 81.5 | 85.1 | 33.1 | 94.3 | 87.6 | 66.9 | 68.3 | 18.5 | 91.9 | 75.2 | 70.0 |
| | | LP++ | 72.9 | 68.0 | 84.1 | 57.2 | 78.0 | 75.4 | 71.3 | 29.7 | 76.6 | 51.8 | 64.7 | 96.3 | 82.1 | 85.5 | 31.6 | 92.5 | 89.3 | 67.8 | 73.2 | 17.7 | 92.9 | 77.5 | 69.8 |
| | | aTLAS | 59.4 | 51.1 | 88.8 | 59.2 | 87.8 | 68.5 | 61.9 | 67.7 | 84.9 | 61.9 | 62.1 | 95.8 | 81.5 | 83.1 | 24.4 | 71.4 | 88.7 | 53.4 | 68.7 | 16.1 | 89.9 | 63.8 | 67.7 |
| | | aTLAS w/ Tip | 72.3 | 67.2 | 81.1 | 83.0 | 94.3 | 80.3 | 68.8 | 28.9 | 75.7 | 77.4 | 65.7 | 97.1 | 84.0 | 89.0 | 40.2 | 94.9 | 89.8 | 71.8 | 77.3 | 20.2 | 95.5 | 81.6 | 74.4 |
| RN101 | 0 | CLIP | 61.0 | 43.6 | 30.7 | 37.7 | 51.4 | 58.5 | 59.5 | 31.5 | 80.8 | 47.7 | 62.3 | 96.8 | 83.0 | 80.9 | 18.5 | 65.3 | 86.9 | 49.6 | 64.5 | 16.9 | 81.8 | 58.5 | 57.6 |
| | 1 | Tip | 66.0 | 50.6 | 60.1 | 41.2 | 54.7 | 63.5 | 63.2 | 39.5 | 80.9 | 52.1 | 63.3 | 96.7 | 84.0 | 84.0 | 22.5 | 77.9 | 87.6 | 53.2 | 65.7 | 17.4 | 87.2 | 67.0 | 62.8 |
| | | LP++ | 64.8 | 55.4 | 65.4 | 43.4 | 56.4 | 66.2 | 62.5 | 30.8 | 82.7 | 54.0 | 63.0 | 96.8 | 83.6 | 83.0 | 22.6 | 79.2 | 86.8 | 51.7 | 65.1 | 17.4 | 86.6 | 68.8 | 63.0 |
| | | aTLAS | 62.5 | 43.6 | 73.9 | 55.3 | 75.9 | 60.4 | 61.1 | 56.3 | 82.6 | 61.4 | 62.9 | 96.8 | 83.6 | 82.7 | 20.1 | 65.3 | 86.9 | 50.7 | 66.8 | 16.9 | 81.8 | 58.5 | 63.9 |
| | | aTLAS w/ Tip | 61.4 | 44.7 | 52.6 | 49.7 | 76.0 | 61.5 | 63.2 | 50.9 | 84.4 | 58.9 | 62.9 | 96.8 | 83.6 | 83.2 | 21.3 | 66.1 | 87.1 | 50.7 | 65.4 | 17.0 | 87.0 | 59.8 | 62.9 |
| | 2 | Tip | 67.3 | 58.4 | 63.0 | 37.5 | 61.9 | 68.0 | 65.6 | 37.0 | 82.5 | 52.5 | 63.8 | 97.0 | 84.0 | 85.0 | 24.1 | 87.4 | 87.7 | 54.8 | 64.4 | 17.6 | 87.9 | 70.8 | 64.5 |
| | | LP++ | 67.5 | 56.4 | 66.4 | 42.5 | 67.7 | 69.8 | 64.8 | 36.4 | 81.0 | 53.7 | 63.4 | 97.1 | 83.7 | 84.5 | 23.9 | 83.8 | 86.5 | 57.0 | 72.1 | 17.6 | 86.6 | 70.0 | 65.1 |
| | | aTLAS | 62.9 | 45.3 | 79.3 | 62.2 | 82.2 | 66.0 | 61.4 | 60.0 | 82.3 | 63.2 | 63.1 | 96.8 | 83.6 | 83.6 | 21.7 | 67.8 | 86.9 | 51.4 | 67.7 | 17.0 | 83.5 | 58.5 | 65.7 |
| | | aTLAS w/ Tip | 66.8 | 51.4 | 65.3 | 63.9 | 85.2 | 59.9 | 64.3 | 54.6 | 83.0 | 60.4 | 63.6 | 96.9 | 83.6 | 83.9 | 22.0 | 66.3 | 87.1 | 52.3 | 65.9 | 17.3 | 88.8 | 63.3 | 65.7 |
| | 4 | Tip | 69.9 | 60.0 | 74.7 | 43.8 | 76.8 | 74.7 | 67.9 | 33.7 | 80.9 | 53.3 | 64.7 | 97.2 | 84.3 | 85.1 | 27.3 | 90.0 | 88.0 | 59.9 | 71.8 | 18.1 | 91.4 | 74.3 | 67.6 |
| | | LP++ | 70.4 | 61.9 | 69.7 | 48.7 | 71.1 | 74.4 | 66.8 | 32.7 | 82.5 | 56.3 | 64.3 | 95.9 | 83.5 | 85.2 | 26.3 | 87.2 | 89.6 | 58.2 | 71.1 | 18.4 | 88.6 | 73.0 | 67.1 |
| | | aTLAS | 64.0 | 47.3 | 80.8 | 63.8 | 80.0 | 70.0 | 62.5 | 59.4 | 87.4 | 64.8 | 63.4 | 97.0 | 83.6 | 83.5 | 21.7 | 68.8 | 87.8 | 51.4 | 67.7 | 17.0 | 87.3 | 58.5 | 66.7 |
| | | aTLAS w/ Tip | 70.9 | 61.5 | 61.8 | 73.1 | 89.9 | 75.2 | 65.9 | 59.9 | 85.2 | 62.9 | 64.9 | 97.1 | 83.8 | 84.5 | 24.3 | 72.1 | 89.0 | 54.5 | 74.6 | 18.0 | 89.9 | 67.5 | 69.4 |
| | 8 | Tip | 73.7 | 66.1 | 79.8 | 54.7 | 77.3 | 77.9 | 69.6 | 34.0 | 82.3 | 56.8 | 65.1 | 97.4 | 84.9 | 86.8 | 31.2 | 93.4 | 88.4 | 67.3 | 71.9 | 18.6 | 91.0 | 76.3 | 70.2 |
| | | LP++ | 71.6 | 64.5 | 78.5 | 54.4 | 81.0 | 77.2 | 69.0 | 30.8 | 83.2 | 57.9 | 65.6 | 96.9 | 84.3 | 86.0 | 28.9 | 88.2 | 89.4 | 62.9 | 73.7 | 19.4 | 90.7 | 76.4 | 69.6 |
| | | aTLAS | 64.9 | 49.5 | 86.5 | 66.5 | 87.6 | 73.1 | 63.2 | 66.6 | 88.1 | 67.0 | 64.0 | 96.9 | 83.6 | 85.3 | 24.4 | 72.8 | 88.2 | 53.6 | 74.1 | 17.1 | 91.3 | 66.1 | 69.4 |
| | | aTLAS w/ Tip | 75.2 | 63.8 | 83.6 | 79.4 | 93.0 | 78.6 | 67.4 | 76.6 | 93.3 | 75.2 | 64.6 | 97.2 | 84.1 | 88.2 | 33.5 | 93.3 | 89.7 | 67.8 | 78.0 | 17.9 | 91.3 | 80.0 | 76.0 |
| | 16 | Tip | 77.6 | 68.3 | 83.1 | 62.8 | 82.0 | 80.5 | 71.8 | 33.1 | 84.9 | 58.3 | 65.8 | 97.8 | 85.0 | 88.0 | 34.5 | 94.3 | 88.6 | 70.5 | 72.1 | 19.9 | 92.3 | 78.3 | 72.3 |
| | | LP++ | 74.8 | 69.2 | 81.7 | 57.5 | 84.8 | 80.1 | 70.1 | 40.4 | 84.8 | 59.6 | 66.9 | 97.2 | 84.8 | 87.6 | 30.6 | 89.5 | 89.6 | 65.1 | 70.7 | 20.7 | 91.3 | 78.0 | 71.6 |
| | | aTLAS | 67.4 | 55.0 | 88.5 | 70.5 | 90.6 | 75.2 | 66.3 | 69.0 | 94.5 | 77.3 | 65.2 | 97.8 | 84.9 | 86.5 | 23.1 | 69.7 | 91.6 | 55.5 | 77.7 | 17.9 | 90.3 | 66.2 | 71.8 |
| | | aTLAS w/ Tip | 80.5 | 64.9 | 88.8 | 84.4 | 93.9 | 84.0 | 72.7 | 73.8 | 92.9 | 78.1 | 65.8 | 97.1 | 84.1 | 89.0 | 41.0 | 94.5 | 91.2 | 72.8 | 76.2 | 20.0 | 94.5 | 80.8 | 78.2 |
| ViT-B/16 | 0 | CLIP | 64.0 | 45.0 | 56.6 | 42.9 | 50.6 | 67.6 | 65.5 | 44.6 | 90.7 | 68.2 | 68.4 | 97.8 | 85.7 | 86.9 | 24.4 | 71.1 | 87.2 | 56.8 | 76.7 | 22.9 | 84.4 | 66.0 | 64.8 |
| | 1 | Tip | 67.7 | 53.8 | 68.7 | 47.8 | 74.2 | 71.3 | 68.8 | 51.9 | 91.6 | 70.0 | 69.1 | 98.3 | 88.8 | 87.6 | 29.8 | 87.0 | 91.8 | 59.5 | 78.3 | 22.9 | 89.2 | 72.9 | 70.0 |
| | | LP++ | 67.8 | 52.8 | 68.1 | 43.1 | 50.6 | 75.6 | 68.0 | 52.3 | 90.7 | 69.0 | 68.5 | 98.2 | 88.9 | 88.2 | 29.6 | 85.3 | 91.4 | 61.3 | 77.6 | 23.0 | 91.4 | 74.6 | 68.9 |
| | | aTLAS | 64.0 | 45.0 | 82.2 | 50.7 | 86.5 | 67.6 | 65.7 | 70.9 | 92.5 | 72.8 | 68.4 | 97.8 | 85.7 | 87.1 | 26.9 | 71.1 | 87.2 | 56.8 | 77.6 | 22.9 | 90.8 | 66.0 | 69.8 |
| | | aTLAS w/ Tip | 66.3 | 51.5 | 82.2 | 64.3 | 92.2 | 71.3 | 69.4 | 76.6 | 90.9 | 75.0 | 68.5 | 98.3 | 88.9 | 88.0 | 29.3 | 83.1 | 89.0 | 59.6 | 76.6 | 23.0 | 88.7 | 70.4 | 72.9 |
| | 2 | Tip | 68.8 | 59.3 | 77.6 | 44.4 | 68.4 | 76.1 | 70.4 | 46.7 | 92.0 | 71.1 | 69.8 | 98.2 | 89.4 | 89.1 | 31.5 | 92.0 | 90.2 | 61.6 | 78.8 | 23.4 | 91.4 | 77.3 | 71.2 |
| | | LP++ | 71.2 | 60.5 | 75.5 | 47.9 | 63.0 | 77.4 | 70.0 | 52.4 | 92.3 | 70.8 | 69.6 | 98.2 | 88.9 | 89.0 | 33.5 | 91.1 | 90.6 | 64.3 | 77.6 | 23.2 | 89.8 | 74.9 | 71.4 |
| | | aTLAS | 66.3 | 45.0 | 84.5 | 64.5 | 92.4 | 68.9 | 67.2 | 68.5 | 92.9 | 74.5 | 69.3 | 97.8 | 86.6 | 88.0 | 26.2 | 73.5 | 88.8 | 56.7 | 77.6 | 22.9 | 90.8 | 66.1 | 71.4 |
| | | aTLAS w/ Tip | 70.5 | 58.6 | 84.3 | 66.8 | 93.6 | 75.5 | 70.9 | 72.8 | 91.6 | 75.4 | 68.8 | 98.4 | 89.0 | 89.2 | 33.6 | 89.3 | 89.4 | 64.2 | 77.8 | 23.2 | 90.8 | 74.9 | 74.9 |
| | 4 | Tip | 73.6 | 60.8 | 77.4 | 51.7 | 78.6 | 82.6 | 72.0 | 45.2 | 92.0 | 71.5 | 70.1 | 98.6 | 89.4 | 90.5 | 36.1 | 95.0 | 91.5 | 68.1 | 78.4 | 23.8 | 91.6 | 78.3 | 73.5 |
| | | LP++ | 75.0 | 62.7 | 79.3 | 55.1 | 81.9 | 82.3 | 73.1 | 50.6 | 91.6 | 72.3 | 70.8 | 98.4 | 89.0 | 90.8 | 35.8 | 93.8 | 91.6 | 67.5 | 81.6 | 23.8 | 91.9 | 78.5 | 74.4 |
| | | aTLAS | 67.2 | 51.4 | 87.8 | 67.6 | 92.7 | 73.0 | 68.2 | 78.3 | 93.1 | 76.7 | 70.0 | 97.8 | 87.4 | 89.0 | 30.2 | 75.7 | 89.6 | 56.8 | 77.6 | 22.9 | 92.6 | 66.4 | 73.3 |
| | | aTLAS w/ Tip | 75.0 | 66.2 | 90.0 | 77.9 | 91.2 | 81.4 | 72.3 | 74.3 | 92.3 | 77.0 | 69.3 | 97.8 | 88.9 | 89.6 | 38.7 | 94.7 | 92.0 | 70.4 | 78.8 | 23.5 | 94.4 | 79.2 | 78.0 |
| | 8 | Tip | 77.8 | 68.6 | 84.6 | 66.2 | 86.8 | 84.1 | 73.9 | 48.3 | 92.9 | 72.5 | 70.6 | 98.3 | 89.1 | 90.7 | 38.8 | 96.2 | 92.6 | 72.9 | 79.2 | 24.5 | 92.6 | 81.7 | 76.5 |
| | | LP++ | 78.4 | 66.9 | 84.0 | 61.4 | 86.5 | 84.6 | 75.0 | 48.0 | 93.1 | 73.1 | 72.1 | 98.5 | 90.0 | 91.2 | 39.4 | 94.8 | 92.0 | 72.5 | 82.0 | 24.4 | 92.7 | 81.0 | 76.5 |
| | | aTLAS | 68.8 | 53.6 | 91.6 | 74.0 | 91.5 | 75.7 | 69.0 | 80.5 | 94.3 | 79.2 | 70.4 | 98.3 | 88.2 | 89.0 | 31.1 | 78.0 | 92.4 | 58.3 | 77.6 | 22.9 | 91.9 | 68.8 | 74.8 |
| | | aTLAS w/ Tip | 79.9 | 69.2 | 92.5 | 84.0 | 94.3 | 85.1 | 73.2 | 76.0 | 94.3 | 78.9 | 69.6 | 98.3 | 89.1 | 90.0 | 43.9 | 96.2 | 92.5 | 75.1 | 80.0 | 23.8 | 90.7 | 82.0 | 79.9 |
| | 16 | Tip | 80.4 | 71.6 | 85.0 | 70.7 | 91.5 | 85.7 | 75.6 | 44.6 | 93.3 | 72.8 | 71.3 | 98.5 | 90.0 | 90.4 | 44.3 | 96.5 | 91.9 | 76.9 | 77.7 | 25.5 | 93.8 | 83.0 | 77.8 |
| | | LP++ | 80.1 | 71.5 | 86.6 | 68.6 | 87.5 | 86.8 | 76.5 | 44.6 | 93.1 | 74.3 | 73.1 | 98.7 | 90.2 | 92.0 | 42.7 | 95.2 | 93.3 | 75.6 | 81.5 | 26.0 | 94.3 | 84.2 | 78.0 |
| | | aTLAS | 69.5 | 54.8 | 92.9 | 76.3 | 92.8 | 78.3 | 69.6 | 79.4 | 95.1 | 80.0 | 70.3 | 98.5 | 88.8 | 89.3 | 30.3 | 78.8 | 92.6 | 59.3 | 79.2 | 23.4 | 91.8 | 71.5 | 75.6 |
| | | aTLAS w/ Tip | 83.0 | 73.5 | 93.4 | 87.3 | 96.5 | 86.7 | 75.1 | 80.6 | 93.7 | 78.3 | 72.2 | 98.5 | 89.1 | 91.8 | 49.8 | 97.1 | 93.1 | 79.3 | 77.6 | 24.5 | 94.9 | 84.6 | 81.8 |
| ViT-L/14 | 0 | CLIP | 79.5 | 54.8 | 61.9 | 50.6 | 77.0 | 73.7 | 68.8 | 51.3 | 95.6 | 76.1 | 75.0 | 99.2 | 89.6 | 90.3 | 32.8 | 77.7 | 93.8 | 63.1 | 78.2 | 32.0 | 85.3 | 74.1 | 71.8 |
| | 1 | Tip | 79.8 | 58.3 | 79.3 | 56.8 | 77.0 | 78.9 | 72.0 | 59.3 | 95.8 | 77.9 | 76.2 | 99.3 | 94.5 | 90.3 | 40.4 | 93.2 | 94.5 | 67.7 | 79.3 | 32.0 | 90.8 | 79.7 | 77.7 |
| | | LP++ | 79.5 | 61.6 | 77.7 | 56.2 | 77.0 | 81.3 | 72.2 | 57.8 | 95.6 | 79.1 | 75.7 | 99.4 | 93.2 | 90.9 | 40.4 | 92.8 | 93.8 | 70.3 | 79.3 | 32.1 | 90.8 | 78.1 | 76.1 |
| | | aTLAS | 79.5 | 57.1 | 81.2 | 70.1 | 90.8 | 76.0 | 69.9 | 75.4 | 95.8 | 81.1 | 75.0 | 99.2 | 90.3 | 90.3 | 36.0 | 81.0 | 93.8 | 63.1 | 78.2 | 32.0 | 91.3 | 74.9 | 77.9 |
| | | aTLAS w/ Tip | 79.5 | 60.6 | 80.7 | 67.7 | 93.6 | 78.3 | 72.5 | 71.5 | 95.6 | 81.9 | 75.3 | 99.4 | 93.2 | 91.9 | 40.2 | 88.9 | 93.8 | 68.3 | 78.5 | 32.0 | 90.8 | 79.0 | 77.9 |
| | 2 | Tip | 80.0 | 63.8 | 84.3 | 60.9 | 77.0 | 81.7 | 74.0 | 58.6 | 95.7 | 78.7 | 76.5 | 99.4 | 93.5 | 92.4 | 43.8 | 95.8 | 93.8 | 71.3 | 79.3 | 32.8 | 92.3 | 81.2 | 77.6 |
| | | LP++ | 81.8 | 64.6 | 83.7 | 59.5 | 77.9 | 81.6 | 74.2 | 58.0 | 96.0 | 79.8 | 75.9 | 99.3 | 93.1 | 93.0 | 42.2 | 95.5 | 93.8 | 73.5 | 79.9 | 32.1 | 92.3 | 81.6 | 78.3 |
| | | aTLAS | 79.5 | 59.2 | 89.7 | 75.2 | 93.9 | 81.0 | 70.5 | 75.3 | 95.7 | 82.9 | 75.8 | 99.2 | 91.2 | 91.6 | 36.8 | 86.0 | 95.0 | 64.2 | 78.3 | 32.0 | 93.0 | 76.9 | 78.3 |
| | | aTLAS w/ Tip | 79.5 | 63.0 | 89.3 | 81.2 | 93.9 | 84.5 | 74.7 | 74.6 | 95.6 | 83.5 | 75.7 | 99.4 | 93.2 | 92.2 | 42.0 | 94.7 | 93.8 | 74.6 | 79.4 | 32.1 | 92.0 | 80.8 | 80.5 |
| | 4 | Tip | 82.3 | 67.8 | 85.1 | 69.1 | 86.7 | 84.7 | 75.8 | 59.8 | 95.9 | 80.0 | 77.1 | 99.4 | 93.1 | 93.7 | 48.3 | 97.7 | 94.9 | 76.1 | 79.4 | 33.4 | 91.9 | 84.3 | 79.8 |
| | | LP++ | 83.2 | 67.2 | 88.0 | 72.6 | 86.8 | 86.2 | 76.6 | 58.3 | 96.2 | 81.1 | 77.3 | 99.5 | 93.0 | 93.8 | 46.8 | 97.0 | 93.8 | 76.7 | 79.3 | 33.4 | 92.5 | 85.1 | 80.1 |
| | | aTLAS | 80.2 | 60.8 | 91.2 | 80.0 | 91.1 | 80.8 | 72.5 | 81.4 | 97.0 | 84.6 | 76.6 | 99.2 | 91.4 | 91.7 | 39.3 | 86.6 | 93.9 | 66.5 | 78.3 | 32.0 | 92.3 | 78.3 | 79.3 |
| | | aTLAS w/ Tip | 83.8 | 68.7 | 92.9 | 84.6 | 94.1 | 85.7 | 76.6 | 76.7 | 96.2 | 83.9 | 76.2 | 99.4 | 93.2 | 93.2 | 48.7 | 96.2 | 94.1 | 76.7 | 79.7 | 32.3 | 94.6 | 85.1 | 82.4 |
| | 8 | Tip | 83.8 | 71.2 | 82.2 | 76.4 | 88.6 | 86.9 | 77.7 | 56.5 | 96.5 | 80.7 | 77.1 | 99.4 | 93.7 | 92.8 | 51.2 | 98.1 | 94.5 | 80.5 | 79.1 | 34.3 | 92.7 | 86.5 | 81.0 |
| | | LP++ | 84.9 | 72.0 | 87.3 | 74.1 | 91.7 | 87.0 | 78.5 | 52.9 | 96.8 | 81.5 | 78.5 | 99.6 | 93.5 | 94.2 | 49.4 | 97.5 | 94.4 | 80.1 | 81.6 | 34.1 | 93.2 | 85.5 | 81.3 |
| | | aTLAS | 80.9 | 62.9 | 91.4 | 82.2 | 94.8 | 83.8 | 72.5 | 82.8 | 97.4 | 85.8 | 76.8 | 99.2 | 92.0 | 92.6 | 42.4 | 89.1 | 94.6 | 68.9 | 80.0 | 32.0 | 92.3 | 80.7 | 80.7 |
| | | aTLAS w/ Tip | 86.1 | 74.0 | 95.5 | 86.6 | 96.8 | 87.8 | 77.7 | 79.6 | 96.7 | 85.9 | 76.4 | 99.2 | 93.4 | 92.9 | 53.8 | 98.3 | 94.1 | 79.7 | 79.8 | 32.7 | 95.5 | 86.8 | 84.1 |
| | 16 | Tip | 87.3 | 73.5 | 88.3 | 74.5 | 90.8 | 89.4 | 78.8 | 56.4 | 96.6 | 81.5 | 78.4 | 99.5 | 93.7 | 94.0 | 56.8 | 98.1 | 93.8 | 82.3 | 80.1 | 35.3 | 95.8 | 88.0 | 82.4 |
| | | LP++ | 87.7 | 76.4 | 90.6 | 75.6 | 91.9 | 89.7 | 80.0 | 56.2 | 97.0 | 82.3 | 79.4 | 99.5 | 94.1 | 94.8 | 52.8 | 97.8 | 94.8 | 82.0 | 83.9 | 34.8 | 95.3 | 88.1 | 83.1 |
| | | aTLAS | 81.4 | 66.9 | 94.0 | 83.9 | 94.4 | 85.7 | 73.7 | 83.4 | 97.9 | 86.4 | 77.4 | 99.2 | 93.0 | 93.0 | 44.2 | 89.2 | 94.7 | 70.9 | 81.5 | 32.5 | 94.6 | 81.3 | 81.8 |
| | | aTLAS w/ Tip | 88.0 | 78.0 | 95.3 | 89.2 | 95.2 | 89.9 | 79.0 | 78.5 | 96.9 | 86.0 | 78.0 | 99.2 | 93.4 | 94.2 | 59.9 | 97.8 | 94.4 | 83.4 | 82.3 | 33.2 | 95.0 | 89.1 | 85.3 |

Table 13: Accuracy of few-shot methods trained on ImageNet [52] and tested on out-of-domain datasets, for $k \in \{4, 16\}$. Results are produced by CLIP with ViT-B/32 backbone and averaged across three random seeds.

| | $k = 4$ | | | | | $k = 16$ | | | | |
|---|---|---|---|---|---|---|---|---|---|---|
| Method | INet | INet-A | INet-R | INet-Sketch | INetV2 | INet | INet-A | INet-R | INet-Sketch | INetV2 |
| Zero-shot | 63.4 | **31.5** | **42.3** | 69.2 | 62.7 | 63.4 | 31.5 | 42.3 | 69.2 | 62.7 |
| aTLAS | 64.7 ± 0.0 | 30.8 ± 0.0 | **42.3** ± 0.0 | **69.9** ± 0.1 | 64.1 ± 0.0 | 65.5 ± 0.0 | **31.7** ± 0.1 | **42.9** ± 0.0 | **70.4** ± 0.0 | 64.7 ± 0.0 |
| Tip-Adapter | 64.9 ± 0.1 | 30.8 ± 0.1 | 41.3 ± 0.2 | 68.5 ± 0.1 | 63.4 ± 0.0 | 66.1 ± 0.0 | 29.1 ± 0.2 | 40.5 ± 0.0 | 67.7 ± 0.1 | 63.7 ± 0.1 |
| LP++ | 65.7 ± 0.0 | 30.1 ± 0.2 | 41.0 ± 0.2 | 67.8 ± 0.1 | 64.4 ± 0.0 | 68.0 ± 0.0 | 29.2 ± 0.1 | 41.0 ± 0.0 | 67.0 ± 0.1 | 66.0 ± 0.0 |
| aTLAS w/ Tip | **66.0** ± 0.0 | 30.2 ± 0.1 | 41.6 ± 0.1 | 69.2 ± 0.2 | 64.5 ± 0.0 | 68.0 ± 0.0 | 29.4 ± 0.3 | 41.4 ± 0.1 | 68.9 ± 0.2 | 65.4 ± 0.1 |
| aTLAS w/ LP++ | **66.0** ± 0.0 | 29.1 ± 0.2 | 41.2 ± 0.2 | 67.9 ± 0.4 | **64.8** ± 0.0 | **68.9** ± 0.0 | 28.7 ± 0.1 | 41.9 ± 0.0 | 67.5 ± 0.1 | **67.0** ± 0.0 |

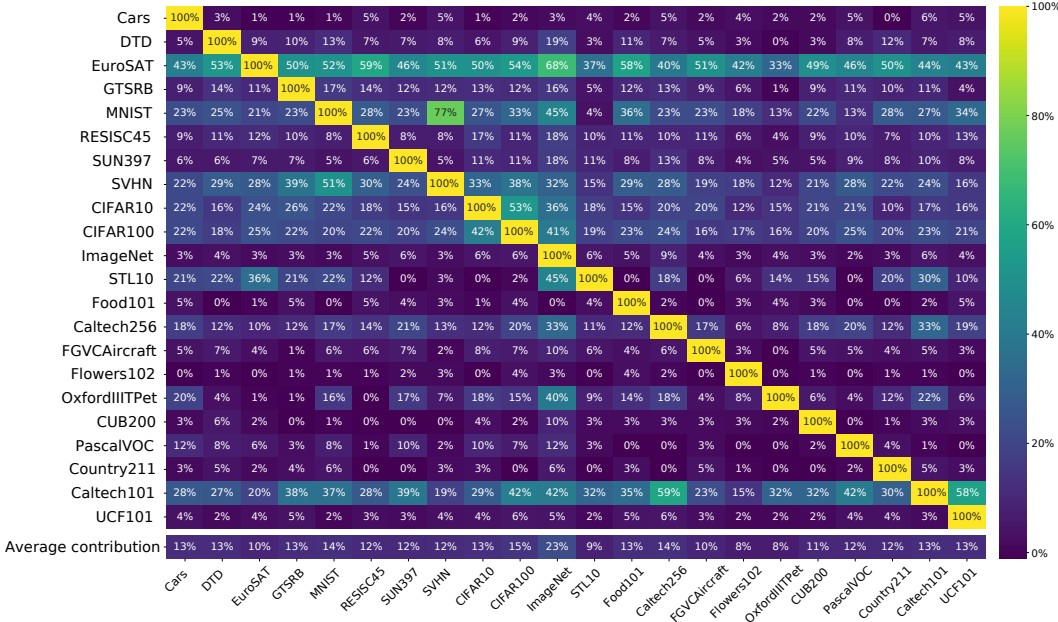

Figure 13: Accuracy improvement of aTLAS (16-shot) using one task vector normalised by that of fine-tuning in the full parameter space (all training data). Each column corresponds to a unique task vector, and reflects the relative improvement it leads to on different target datasets. Each row reflects the relative improvement on a dataset, using different task vectors.

## D.4 Out-of-domain generalisation

We show detailed results for out-of-domain generalisation over $k \in \{4, 16\}$ shots in Table 13. These results correspond to those presented in Figure 5c. aTLAS is the only method that consistently improves test accuracy over the zero-shot model on out-of-domain images. When combined with LP++ or Tip-Adapter, aTLAS can be observed to improve the out-of-domain generalisation of these methods.

## D.5 Relative significance of individual task vectors

In this section, we examine the informativeness of a task vector across different target datasets. To this end, we apply aTLAS to each of the 22 datasets using only one task vector. For each dataset, we compute the relative accuracy improvement, that is, the accuracy improvement of aTLAS normalised by that of fine-tuning in the full parameter space. Note that aTLAS is applied under the 16-shot setting, while standard fine-tuning uses all training data available. Results are shown in Figure 13. We first note that certain datasets are more prone to accuracy improvement, such as EuroSAT, MNIST, etc., as indicated by the high percentage across entire rows. This is most likely due to the low intrinsic dimensionality of the task. In addition, we highlight the average improvement in the last row. Notably, certain task vectors, e.g., ImageNet task vector, are particularly informative while others, such as those from Flowers102 and OxfordPets are much less so. These results illustrate the varying contributions different task vectors can have depending on the target dataset, which also motivated subsequent efforts on careful task vector selection.

Table 14: Few-shot accuracy when only using a budget of $b$ task vectors with different selection strategies. We report results for 4 and 16 shots. The results are averaged over 22 datasets and three random seeds. CLIP with ViT-B/32 backbone is used. Highest performance in each section is highlighted in bold.

| Shots ($k$) | Strategy | $b = 0$ | $b = 1$ | $b = 2$ | $b = 5$ | $b = 10$ | $b = 15$ | $b = 21$ |
|---|---|---|---|---|---|---|---|---|
| 4 | Random | 60.39 | $63.5 \pm 0.0$ | $64.8 \pm 0.1$ | $67.1 \pm 0.2$ | $69.3 \pm 0.1$ | $69.7 \pm 0.1$ | $70.0 \pm 0.0$ |
| | Features | 60.39 | $65.6 \pm 0.1$ | $67.6 \pm 0.2$ | $68.8 \pm 0.1$ | $\mathbf{69.5} \pm 0.1$ | $69.8 \pm 0.1$ | $70.0 \pm 0.0$ |
| | Grad. whole | 60.39 | $64.4 \pm 0.1$ | $65.8 \pm 0.2$ | $67.2 \pm 0.1$ | $69.1 \pm 0.1$ | $69.6 \pm 0.1$ | $70.0 \pm 0.0$ |
| | Grad. blockwise | 60.39 | $\mathbf{67.3} \pm 0.2$ | $\mathbf{68.2} \pm 0.0$ | $\mathbf{68.9} \pm 0.1$ | $69.1 \pm 0.2$ | $\mathbf{69.7} \pm 0.2$ | $70.0 \pm 0.0$ |
| 16 | Random | 60.39 | $64.7 \pm 0.1$ | $66.0 \pm 0.1$ | $68.8 \pm 0.1$ | $70.8 \pm 0.0$ | $72.0 \pm 0.1$ | $72.8 \pm 0.1$ |
| | Features | 60.39 | $66.2 \pm 0.0$ | $68.1 \pm 0.2$ | $70.3 \pm 0.0$ | $\mathbf{71.7} \pm 0.1$ | $\mathbf{72.4} \pm 0.1$ | $72.8 \pm 0.1$ |
| | Grad. whole | 60.39 | $65.2 \pm 0.1$ | $66.2 \pm 0.1$ | $68.3 \pm 0.2$ | $71.5 \pm 0.1$ | $72.2 \pm 0.1$ | $72.8 \pm 0.1$ |
| | Grad. blockwise | 60.39 | $\mathbf{68.3} \pm 0.1$ | $\mathbf{69.3} \pm 0.1$ | $\mathbf{70.5} \pm 0.1$ | $71.6 \pm 0.0$ | $72.3 \pm 0.0$ | $72.8 \pm 0.1$ |

## D.6 Task vector budget and selection

In this section, we provide details for selecting a budget of $b$ task vectors with feature-based and gradient-based strategies, as introduced in Section 5.2.

**Feature based selection**. For each dataset $\mathcal{D}_i$, we compute the average image representation $\bar{\mathbf{z}}_i$ of the dataset using the zero-shot model as follows

$$\bar{\mathbf{z}}_i = \mathbf{E}_{\mathbf{x} \in \mathcal{D}_i}[f(\mathbf{x}; \boldsymbol{\theta}_0)]. \tag{13}$$

Given a target dataset $\mathcal{D}_t$, we simply compute the cosine similarity between its feature representation $\bar{\mathbf{z}}_t$ and that of each other dataset $\bar{\mathbf{z}}_i, i \neq t$. Subsequently, $b$ task vectors corresponding to the datasets with highest similarity will be selected.

**Gradient-based selection**.

Given a target dataset $\mathcal{D}_t$, we may directly compute the gradient with respect to the $m$ learnable coefficients for each of the $n$ task vectors. However, as one important motivation behind task vector selection is to reduce memory consumption, using all $n$ task vectors to compute the gradient defeats the purpose. Therefore, we instead only load a group of $b$ task vectors ($b < n$), compute the gradient with respect to their learnable coefficients, and repeat for other groups. With this sequential computation, the gradient across different groups is not calibrated. Nevertheless, we empirically found this strategy to work well. Denote the partial derivative of the loss on dataset $\mathcal{D}_t$ with respective to a learnable coefficient $\lambda_i^{(j)}$ by $\dot{\lambda}_i^{(j)}$, such that

$$\dot{\lambda}_i^{(j)} = \mathbf{E}_{(\mathbf{x}, \mathbf{y}) \in \mathcal{D}_t} \left[ \frac{\partial \mathcal{L}\left( f\left(\mathbf{x}; \boldsymbol{\theta}_0 + \sum_{i=1}^b \Lambda_i \boldsymbol{\tau}_i\right), \mathbf{y}\right)}{\partial \lambda_i^{(j)}} \right]. \tag{14}$$

For the $i$-th task vector, we may compute its $L_1$ gradient norm, i.e., $\left\| \dot{\lambda}_i^{(1)}, \ldots, \dot{\lambda}_i^{(m)} \right\|_1$, and select task vectors with larger gradient. Alternatively, we may select task vectors block by block. Specifically, for the $j$-th parameter block, we inspect the absolute values of the partial derivatives for the corresponding coefficients, i.e., $\left| \dot{\lambda}_i^{(j)} \right|$, and select task vectors with higher absolute values. This process is repeated for each parameter block, thus allowing different parameter blocks to have different selections. Crucially, for low budgets, particularly $b = 1$, this enables our method to effectively exploit more task vectors than the budget specifies. The impact of this can be observed in Table 14 (corresponding to Figure 6), that blockwise selection significantly outperforms other methods when the budget is low.

## D.7 LoRAs as task vectors

We fine-tune LoRAs for ViT-B/32 using the LoRA-Torch [36] library with ranks 4, 16 and 64. We stop at rank 64 as we do not observe improvements beyond it. We train LoRAs on attention and MLP layers and use the same settings as for full finetuning but with a learning rate of $10^{-3}$.

Table 15 shows additional results using LoRAs as task vectors. We study learning the effect of fine-tuning the LoRAs task vectors on attention layers only (as done in the original LoRA paper [23]) or on the MLPs. Although the original LoRA paper recommendeds training on the attention layers only [23], we observe that training on MLP layers is important to produce strong LoRA task vectors.

Table 15: Additional few-shot recognition results using LoRAs trained on attention layers, MLP layers or both. Results are averaged across 22 datasets over three seeds, with $\times 1$ standard deviation. Rank 16 is used for LoRAs.

| Task vector type | Method | 0-shot | 1-shot | 2-shot | 4-shot | 8-shot | 16-shot |
|---|---|---|---|---|---|---|---|
| LoRAs (Attn.) | aTLAS | 60.4 | $63.5 \pm 0.1$ | $65.1 \pm 0.1$ | $66.6 \pm 0.1$ | $67.9 \pm 0.1$ | $69.5 \pm 0.0$ |
| LoRAs (MLP) | aTLAS | 60.4 | $63.8 \pm 0.1$ | $66.2 \pm 0.1$ | $68.3 \pm 0.1$ | $\mathbf{70.5} \pm 0.1$ | $71.4 \pm 0.0$ |
| LoRAs (Attn. & MLP) | aTLAS | 60.4 | $\mathbf{64.6} \pm 0.1$ | $\mathbf{66.6} \pm 0.2$ | $\mathbf{68.7} \pm 0.1$ | $70.4 \pm 0.1$ | $\mathbf{71.8} \pm 0.1$ |
| LoRAs (Attn. & MLP) | aTLAS w/ LP++ | 60.4 | $67.1 \pm 0.3$ | $70.9 \pm 0.1$ | $73.4 \pm 0.1$ | $75.9 \pm 0.1$ | $78.2 \pm 0.1$ |
| LoRAs (Attn. & MLP) | aTLAS w/ Tip-Adapter | 60.4 | $67.5 \pm 0.1$ | $70.0 \pm 0.1$ | $72.4 \pm 0.1$ | $74.9 \pm 0.1$ | $77.0 \pm 0.1$ |
| Standard | aTLAS w/ LP++ | 60.4 | $\mathbf{68.9} \pm 0.2$ | $\mathbf{71.7} \pm 0.1$ | $74.1 \pm 0.1$ | $75.8 \pm 0.1$ | $77.9 \pm 0.0$ |
| Standard | aTLAS w/ Tip-Adapter | 60.4 | $68.6 \pm 0.4$ | $71.6 \pm 0.2$ | $\mathbf{74.3} \pm 0.1$ | $\mathbf{76.4} \pm 0.1$ | $\mathbf{78.2} \pm 0.0$ |

Table 16: Few-shot recognition performance with gradient-free optimisation. Results are averaged accuracy over 22 datasets, with $1\times$ standard error over 3 random seeds.

| Scaling | Use gradient | Memory (GB) | 0-shot | 1-shot | 2-shot | 4-shot | 8-shot | 16-shot |
|---|---|---|---|---|---|---|---|---|
| Anisotropic | Yes | 10 | 60.4 | $66.7 \pm 0.23$ | $68.3 \pm 0.28$ | $70.0 \pm 0.01$ | $71.7 \pm 0.11$ | $72.8 \pm 0.08$ |
| Isotropic | No | 4 | 60.4 | $\mathbf{63.1} \pm 0.45$ | $\mathbf{64.2} \pm 0.35$ | $\mathbf{65.0} \pm 0.12$ | $\mathbf{65.7} \pm 0.05$ | $\mathbf{65.4} \pm 0.14$ |
| Anisotropic | No | 4 | 60.4 | $61.3 \pm 0.08$ | $61.5 \pm 0.04$ | $61.5 \pm 0.04$ | $61.6 \pm 0.03$ | $61.6 \pm 0.02$ |

## D.8 Gradient-free optimisation

An alternative to save memory during training is to utilise gradient-free methods to learn the coefficients. We follow previous work on the combination of LoRAs [24] and use the nevergrad [49] library. We observe a memory usage reduction of 60% from 10GB to 4GB calculated using a dedicated pytorch function[6]. Results for few-shot recognition are summarised in Table 16. We show that although gradient-free optimisation improves upon the zero-shot model, the performance quickly plateaus as the amount of data increases. In addition, learning anisotropic scaling results in worse performance, most likely due to the relatively high number of parameters.

---

[6]https://pytorch.org/docs/stable/generated/torch.cuda.memory_allocated.html

Table 17: Accuracy after fine-tuning on different percentage of training data for variants of aTLAS $\times K$ and LoRAs [23]. Results are averaged across 22 datasets. Highest accuracy in each section is highlighted in bold.

| Method | # Params | 1% | 5% | 10% | 25% | 35% | 50% | 100% |
|---|---|---|---|---|---|---|---|---|
| aTLAS | 2k | 68.5 | 71.5 | 72.6 | 73.6 | 74.6 | 75.4 | 76.4 |
| aTLAS $\times 5$ | 10k | 69.3 | 72.9 | 74.7 | 76.2 | 76.8 | 77.5 | 78.8 |
| aTLAS $\times 20$ | 40k | 69.5 | 74.0 | 75.6 | 77.5 | 78.2 | 78.9 | 80.5 |
| aTLAS $\times 80$ | 160k | 70.2 | 74.7 | 76.2 | 77.9 | 78.9 | 80.0 | 82.0 |
| aTLAS $\times 1200$ | 2.4M | **71.3** | **75.0** | **76.6** | **78.3** | **80.2** | **81.5** | **83.9** |
| LoRA (rank=16) | 2.4M | 68.8 | 74.1 | 75.6 | 76.8 | 79.0 | 80.6 | 83.6 |

## E  Unsupervised FixMatch

We provide more details on the Unsupervised FixMatch (UFM) approach in this section. FixMatch [54] utilises a labelled set to guide training, which is given as part of the semi-supervised learning protocol, while we produce a class-balanced "labelled" set from unlabelled images. Given a target dataset $\mathcal{D}_t$ consisting of $N$ unlabelled images, we first rank the examples by the prediction scores from the zero-shot model across $C$ classes. We then select the top $\min(N/C, 100)$ examples, that is, at most $100$ examples per class, as a trusted set in absence of a labelled set. The standard cross-entropy loss is applied to the trusted set. For the rest of the unlabelled images, we use a weakly augmented (Open-CLIP [26] validation augmentations) view of an image to produce pseudo-labels, and incur a loss on the strongly augmented view (Tip-Adapter [69] augmentations). Denote an image with weak augmentation by $\mathbf{x}$, its strongly augmented view by $\mathbf{x}'$, and the predictions made by network by $\hat{\mathbf{y}}$ and $\hat{\mathbf{y}}'$, respectively, the unsupervised loss can be expressed as

$$\ell_u(\hat{\mathbf{y}}, \hat{\mathbf{y}}') = -\mathbb{1}(\max(\sigma(\hat{\mathbf{y}})) > \omega)\, \sigma(\hat{\mathbf{y}})^\mathsf{T} \log(\hat{\mathbf{y}}'), \tag{15}$$

$$\sigma(\hat{\mathbf{y}}) = \frac{\hat{\mathbf{y}}^{0.5}}{\mathbf{1}^\mathsf{T}\hat{\mathbf{y}}^{0.5}}, \tag{16}$$

where $\mathbb{1}(\cdot)$ denotes the indicator function, $\sigma(\cdot)$ performs re-normalisation with adjusted temperature scaling, and $\omega$ is a confidence threshold that is linearly adjusted from 0.9 to 1 during training. The trusted set is re-estimated at the beginning of each epoch to account for the improving accuracy of the model. In training, images in the trusted set are over-sampled to constitute one fourth of each batch, as this practice prevents the model from diverging due to confirmation bias [2, 54].

## F  Details of aTLAS $\times K$ variants

Dividing a parameter block into $K$ random partitions allows us to introduce more learnable coefficients to each block, thus scaling up our method flexibly. One draw back of this approach, however, is that masks for the partitions have to be stored in memory, resulting in a linear memory increase with respect to the size of the parameter block and the value $K$. To reduce the memory consumption the of aTLAS $\times K$ variants, we only apply it to LoRAs task vectors. Nevertheless, these memory requirements could most likely be reduced by exploiting sparse matrices or memory efficient matrix indexing techniques, which we plan to investigate in the future.

