# OpenReview forum: "Knowledge Composition using Task Vectors with Learned Anisotropic Scaling"
_NeurIPS.cc/2024/Conference — NeurIPS 2024 poster_

### Official Review · Reviewer_TrF6 · 2024-07-11

**Soundness:** 3
**Presentation:** 3
**Contribution:** 3
**Rating:** 5
**Confidence:** 4

**Summary:**

This paper presents a method called aTLAS, which leverages task vectors to enhance transfer learning in neural networks. Task vectors represent the difference in weights between a pre-trained model and its fine-tuned variant. aTLAS introduces anisotropic scaling to these task vectors by learning different coefficients for each parameter block, which allows for more effective composition of knowledge from different tasks. The method is tested in various scenarios, including task arithmetic, few-shot recognition, and test-time adaptation, demonstrating improvements in performance and efficiency.

**Strengths:**

- An interesting way of using task vectors: the paper extends the concept of task vectors by introducing anisotropic scaling, which enhances the flexibility and effectiveness of knowledge transfer. This approach allows for fine-grained control over the composition of different task vectors, leading to better performance in multi-task settings.

- Evaluation and analysis: the method is evaluated across multiple tasks, including task arithmetic, few-shot learning, and test-time adaptation. The comprehensive set of experiments provides strong evidence for the effectiveness of the proposed approach. It provides insights on the behavior and importance of various parts of the neural network during task vector composition.

- Parameter efficiency: aTLAS is shown to be a parameter-efficient method for fine-tuning, which is particularly valuable in scenarios with limited data. The ability to achieve high performance with fewer learnable parameters is a significant advantage for practical applications.

**Weaknesses:**

- **Only CLIP?** the method is primarily tested on the CLIP model and might not directly generalize to other architectures without significant modifications. Future work should explore the applicability of aTLAS across a broader range of model architectures.

- **Computational complexity (is this scalable)?**  While the method is parameter-efficient, the computation of task vector compositions during training can still be resource-intensive, especially for large models. Strategies to optimize this process or reduce its computational footprint would be beneficial.  The method's scalability with an increasing number of task vectors is not fully explored. While the paper shows that performance improves with more task vectors, it is unclear how this scales with very large sets of task vectors or in more complex multi-task environments. Also, earning different coefficients for each parameter block introduces additional complexity. The benefits of this complexity should be weighed against the potential for simpler approaches that might achieve similar results with less computational overhead.

- **How exactly should one select the task vectors?** The selection of task vectors for composition can significantly impact performance. The paper discusses various selection strategies but does not provide a definitive approach. Further research into more sophisticated selection mechanisms could enhance the method's robustness and effectiveness.


- **How to leverage this in real world?** The experiments are conducted in controlled settings with well-defined datasets. Evaluating the method's performance in more diverse and real-world scenarios would provide a better understanding of its practical applicability and limitations.

**Questions:**

See weaknesses section

**Limitations:**

See weaknesses section

---

> ### Author Rebuttal · Authors · 2024-08-07
>
> We are encouraged that the reviewer find our method interesting and the experiments comprehensive and insightful. We are thankful for the feedbacks. Below, we address the questions and concerns.
>
> **W1. Applicability across different model architectures**
>
> For task arithmetic, we included results with ViT-{B/32, B/16, L/14} backbones (Table 1, 2) following previous practice (Ilharco et al., 2023 and Ortiz-Jimenez et al., 2023). For transfer learning applications including few-shot adaptation and test-time adaptation, we additionally showed results with ResNet-{50, 101} backbones besides ViTs (Table 13). Beyond CLIP models, Ilharco et al. (2023) showed that the properties of task vectors also apply to GPT-2 and T5 models. We believe task vector composition can also be applied to these models and plan to investigate in future work.
>
> **W2. Scalability**
>
> Research on larger pools of task vectors remains part of the future work. In such scenarios, LoRA task vectors emerge as the current optimal solution for mitigating memory usage. As shown in Fig. 6a and Table 14 in Appendix F, LoRA task vectors substantially reduce memory requirements while maintaining an accuracy score comparable to full task vectors, demonstrating their feasibility and potential for large number of task vector. This finding underscores the potential of LoRA task vectors as a memory-efficient solution for exploiting extensive collections of task vectors.
>
> **W3. Definitive approach on task vector selection**
>
> Thank you for pointing this out. We find in Fig. 5 that block-wise gradient-based selection is the best strategy, especially with low budget on task vectors. Therefore, it is our recommended approach and have been explicitly stated at L213 in our revision.
>
> **W4. Applying aTLAS to the real world**
>
> We have demonstrated strong results of our method in five applications. In particular, few-shot adaptation, parameter-efficient fine-tuning, etc., have direct use cases in real-world scenarios. We also conducted experiments across 22 datasets that cover a wide range of domains, in order to test its general practicality. We believe the observations we made on these datasets can reasonably reflect the challenges in the real world.
>
> References:
> - Ilharco et al. Editing models with task arithmetic. ICLR'23
> - Ortiz-Jimenez et al. Task arithmetic in the tangent space: Improved editing of pre-trained models. NeurIPS'23.

---

### Official Review · Reviewer_GVMe · 2024-07-11

**Soundness:** 3
**Presentation:** 3
**Contribution:** 3
**Rating:** 7
**Confidence:** 4

**Summary:**

The paper introduces a method named aTLAS, which leverages task vectors and anisotropic scaling to enhance knowledge composition and transfer in pre-trained models. The authors investigate whether components of task vectors, particularly parameter blocks, exhibit similar characteristics and how these can be used to improve knowledge composition and transfer. The effectiveness of the proposed method is demonstrated in various tasks such as task arithmetic, few-shot recognition, and test-time adaptation.

**Strengths:**

- The introduction of anisotropic scaling at the task vector level is novel and offers higher controllability in model behavior, particularly for task addition and negation.
- The method is thoroughly validated across multiple tasks and datasets, showing significant improvements in performance.
- aTLAS demonstrates strong parameter efficiency, making it suitable for scenarios with limited data.
- The method complements existing few-shot adaptation techniques, leading to additional improvements in performance when combined.

**Weaknesses:**

- Knowledge composition and transfer are limited to the specific pre-trained model architecture, which may restrict its applicability across diverse model architectures.

**Questions:**

- How does the method handle potential conflicts when combining task vectors from very dissimilar domains? Will it negatively affect performance?

**Limitations:**

- Knowledge composition and transfer are limited to the specific pre-trained model architecture, which may restrict its applicability across diverse model architectures.

---

> ### Author Rebuttal · Authors · 2024-08-07
>
> We appreciate the reviewer for recognizing the novelty, parameter efficiency and thorough experimentation in our work, and we are thankful for the feedbacks. We now address the questions and concerns as follows.
>
> **W1. Knowledge composition and transfer are limited to specific pre-trained model architecture.**
>
> While the experiments primarily use ViT-B/32 as the visual encoder, we showed that the proposed method works consistently across other architectures including ViT-B/16, ViT-L/14 (Table 1, 2), ResNet50 and ResNet101 (Table 13). Nevertheless, we acknowledge that aTLAS cannot combine task vectors obtained from different architectures, or transfer the knowledge in task vectors to a different architecture. This may require finding appropriate projections, and remains part of the future work.
>
> **Q1. Task vectors from dissimilar domains.**
>
> We studied the potential conflicts between task vectors using disentanglement error (Ortiz-Jimenez et al., 2023), as shown in Figure 3 of the paper. Specifically, Each row reflects the percentage of data in the corresponding dataset that have altered predictions after combining two task vectors. Therefore, low disentanglement error indicates low interference between task vectors. More importantly, this interference also depends on the dataset. For two datasets A and B, combining their corresponding task vectors into one model may hurt the performance on one dataset more than the other.
>
> Our method, particularly with standard task vectors (Figure 3c), generally decreases the disentanglement error across all pairs, which very effectively reduces the conflict between task vectors, whether they are obtained from similar or dissimilar domains.
>
> References:
> - Ortiz-Jimenez et al. Task arithmetic in the tangent space: Improved editing of pre-trained models. NeurIPS'23.

---

### Official Review · Reviewer_RiZ9 · 2024-07-12

**Soundness:** 3
**Presentation:** 3
**Contribution:** 3
**Rating:** 7
**Confidence:** 4

**Summary:**

The paper enhances the performance of task arithmetic, a recent model editing technique based on weight interpolation, in vision-language models. Instead of the original task- and parameter-independent scaling coefficients of the task vectors, it proposes to learn anisotropic scaling coefficients from validation data, resulting in significant performance improvements, particularly in task addition. The method also proves effective in few-shot learning and test-time adaptation scenarios and as a parameter-efficient fine-tuning (PEFT) method in low-data regimes.

**Strengths:**

- **Originality:** While some parts of the method are not entirely novel (see Weaknesses section), overall, aTLAS goes beyond previous work. The few-shot adaptation application and its relation to PEFT (e.g., using LoRAs as task vectors) are also original contributions to the field of model editing/merging.
- **Quality:** The work is generally of good quality.
- **Significance:** Editing foundation models is an emerging field with promising real-world impact. The results obtained with the proposed method are very good. Specifically, the authors merge the parameters of 8 ViT-L-14 CLIP models while retaining 97.07% of the performance of the single models (see Tab. 2).

**Weaknesses:**

**Originality and references to previous work**

1. The idea of learning task-wise and layer-wise scaling coefficients for the task vectors is one of the main features of Adaptive Model Merging (*AdaMerging*, Yang et al., 2024). This work, which is not currently referenced, must be duly cited, along with a detailed discussion of the similarities and differences between aTLAS and AdaMerging.
2. Similarly, the idea of using test-time adaptation techniques such as entropy optimization was also present in Yang et al. (2024).

**Clarity and missing details**

3. The paper does not report the essential methodological details of aTLAS. Specifically, the loss, optimizer, learning rate, and hyperparameters for learning the scaling coefficients are missing.
4. The disentanglement error (line 143) must be defined and briefly explained. Moreover, as it seems to slightly differ from the original metric defined in Ortiz et al. (2024) – as likely it is computed only for the best set of scaling coefficients – this difference should also be mentioned.
5. In lines 153-155, some citations could be added, e.g., to the fact that the representation built by neural networks is often hierarchical and increases in complexity with the layer depth.
6. The experiments with ResNet backbones are never mentioned in the main text.
7. A comparison of the computational costs compared to standard task arithmetic is missing and should be provided.

Yang, E., Wang, Z., Shen, L., Liu, S., Guo, G., Wang, X. and Tao, D., 2024. AdaMerging: Adaptive model merging for multi-task learning. The Twelfth International Conference on Learning Representations.

Ortiz-Jimenez, G., Favero, A. and Frossard, P., 2024. Task arithmetic in the tangent space: Improved editing of pre-trained models. Advances in Neural Information Processing Systems, 36.

**Questions:**

8. **Multi-task learning vs. learning the coefficients.** Differently from the original task arithmetic technique, aTLAS involves learning the optimal merging coefficients for each task and block from data. How would the performance of fine-tuning the pre-trained model on the same number of data and for the same number of steps compare with your task addition results? What about the accuracy of the model on a control dataset that was not used during fine-tuning, e.g., ImageNet?

**Limitations:**

The paper adequately discusses its limitations. I do not foresee any potential negative societal impacts arising from this study.

---

> ### Author Rebuttal · Authors · 2024-08-07
>
> We appreciate the reviewer's recognition of the novelty, quality and impact of our work, and we are thankful for the feedbacks. In what follows, we now address the questions and concerns.
>
> **W1. Comparison to AdaMerging (Yang et al., 2024)**
>
> We thank the reviewer for suggesting this comparison, and have added this in our revision (Section 3, Tables 2 and 3) with appropriate citations. In short, the idea of anisotropic scaling is a more general formulation, while layer-wise scaling in AdaMerging is a specific variant. Besides, AdaMerging is designed for model merging, similar to task addition, which is one of the five applications we investigated. Below, we detail the key differences between our work and AdaMerging.
> - **Formulation**. We formulated anisotropic scaling as $\Lambda \mathbf{\tau}$, where $\mathbf{\tau}$ denotes a task vector and $\Lambda$ is a block-diagonal scaling matrix. When parameters in the same layer share one scaling coefficient, this formulation specializes to AdaMerging. However, for complex use cases such as parameter-efficient fine-tuning (Sec. 6.2 and Appendix F), we show that scaling different rows, columns or random partitions of weight matrices differently is necessary to increase the representation power. This variant can also be captured by our formulation. In addition, AdaMerging constrains the learned coefficients to be in $[0,1]$ while aTLAS does not. We observe that larger or negative coefficients can be beneficial for task addition in Fig. 10. **The formulation of anisotropic scaling therefore covers a wider range of use cases**.
> - **Task vector variants**. We additionally studied combining linearized task vectors (Ortiz-Jimenez et al., 2023) in Section 4 and Appendix D.3, and using LoRAs as sparse task vectors to reduce memory consumption (Section 6.1).
> - **Training objectives**. AdaMerging uses entropy as an unsupervised objective, while we experimented with constrastive, regularized entropy and pseudo-labelling objectives in Section 5.3. Note that our unsupervised pseudo-labeling algorithm UFM outperforms the entropy objective for test-time adaptation (Table 3).
> - **Applications**.  AdaMerging focuses on model merging, whereas our work investigates additional applications such as task negation (Sec. 4.1), few-shot recognition (Sec 5.1), test-time adaptation (Sec 5.3) and parameter-efficient fine-tuning (Sec 6.2), which, as noted by the reviewer, are original contributions to the field.
> - **Novel insights for model merging**. As discussed in L141-146, anisotropic scaling reduces interference when merging task vectors (Fig. 3), a novel insight uncovered in our paper, highlighting its advantage over isotropic scaling and negating the need for linearized task vectors.
>
> Last, we kindly note that AdaMerging's publication in ICLR'24 coincides with the submission period of NeurIPS'24. Nevertheless, we have added comparison and acknowledgements where appropriate throughout the paper and thank the reviewer for the suggestion.
>
> **W2. Entropy optimisation in AdaMerging**
>
> We thank the reviewer for pointing this out and have added a discussion in L224 to acknowledge this. We highlight that UFM, the pseudo-labelling algorithm we designed, outperforms (regularized) entropy optimisation (SAR, Table 3). AdaMerging has also been included as a baseline in this table.
>
> **W3. Technical details**
>
> They can be found in Appendix A for task addition or negation; Appendix D.1 and D.2 for few-shot learning. In our revision, we have added hyperlinks at L160 for clarity.
>
> **W4. Disentanglement Error**
>
> The reviewer is correct that the disentanglement error is evaluated at the optimal coefficients. We have added detailed mathematical formulas in our revision for clarity, as follows
>
> $\xi(\tau_1, \tau_2) = \textbf{E}_{\mathbf{x} \in \mathcal{D}_1} \big[ \delta \big( f(\mathbf{x}; \mathbf{\theta}_0 + \Lambda^\star_1 \mathbf{\tau}_1), f(\mathbf{x}; \mathbf{\theta}_0 + \Lambda^\star_1 \mathbf{\tau}_1 + \Lambda^\star_2 \mathbf{\tau}_2) \big) \big]$,
>
> where $\Lambda^\star$ denotes the optimal coefficients and $\delta(x_1, x_2)$ is a distance function that returns 1 if $x_1 \neq x_2$ and 0 otherwise.
>
> **W5. Citations**
>
> We have added references to Yosinski et al. (2014) and Kornblith et al. (2019) in our revision.
>
> **W6. ResNet Experiments**
>
> We have added a hyperlink to the results with ResNets (Table 13) in our revision.
>
> **W7. Computational cost**
>
> Standard task addition performs hyper-parameter search on one scaling coefficient ranging from 0 to 1, with an interval of 0.05, and therefore runs inference 21 times. Our method is trained for 10 epochs. Using an RTX 4090, our method takes 12min to train while the hyper-parameter search takes 20min.
>
> **Q1a. Comparison against multi-task fine-tuning**
>
> As shown in figure in the attached PDF, our method consistently outperforms multi-task fine-tuning using different percentage of the validation data, since the training data is considered unavailable. However, with more data, fine-tuning is expected to achieve better performance.
>
> **Q1b. Performance on a control dataset**
>
> With our method, the merged model achieves 58.1% accuracy on ImageNet, which retains 91.6% of the zero-shot accuracy 63.4%. In comparison, the model from multi-task fine-tuning achieves 57.3% accuracy on ImageNet, which shows our method generalizes better.
>
> References:
> - Yang et al.  AdaMerging: Adaptive model merging for multi-task learning. ICLR'24
> - Ortiz-Jimenez et al. Task arithmetic in the tangent space: Improved editing of pre-trained models. NeurIPS'23.
> - Yosinski et al. How transferable are features in deep neural networks? NeurIPS'14.
> - Kornblith et al. Similarity of neural network representations revisited. ICML'19.

---

> > ### Comment · Reviewer_RiZ9 · 2024-08-13
> >
> > Thank you for your detailed response, which successfully addressed all of my concerns. I encourage you to include the additional discussions in the revised version of your submission. Overall, I believe this paper merits acceptance, and I have adjusted my initial score accordingly. Best wishes.

---

### Author Rebuttal · Authors · 2024-08-07

We would like to thank each reviewer for dedicating their time to reviewing our paper. We are encouraged that the reviewers find our work novel/original (Reviewers [RiZ9](https://openreview.net/forum?id=G9OJUgKo4B&noteId=PnDeqZQ2DM), [GVMe](https://openreview.net/forum?id=G9OJUgKo4B&noteId=PnDeqZQ2DM)), our experiment results thorough (Reviewers [GVMe](https://openreview.net/forum?id=G9OJUgKo4B&noteId=PnDeqZQ2DM), [TrF6](https://openreview.net/forum?id=G9OJUgKo4B&noteId=UDzo9FVMVM)) and impactful ([RiZ9](https://openreview.net/forum?id=G9OJUgKo4B&noteId=PnDeqZQ2DM)). In particular, reviewer [RiZ9](https://openreview.net/forum?id=G9OJUgKo4B&noteId=PnDeqZQ2DM) appreciates our contribution in applications such as few-shot adaptation and parameter-efficient fine-tuning, which goes beyond previous work. Reviewers [GVMe](https://openreview.net/forum?id=G9OJUgKo4B&noteId=PnDeqZQ2DM) and [TrF6](https://openreview.net/forum?id=G9OJUgKo4B&noteId=UDzo9FVMVM) appreciate that our experiments provide insights on the behaviors of task vector compositions, such as our method being complementary to existing few-shot methods.

In response to the question from reviewer [RiZ9](https://openreview.net/forum?id=G9OJUgKo4B&noteId=PnDeqZQ2DM), we ran experiments to compare our method aTLAS against multi-task fine-tuning, and have included the new results in the attached PDF. Specifically, we show that our method outperforms multi-task fine-tuning significantly and consistently when using different percentage of data. The validation sets are used in this case following previous practice by Ilharco et al. (2023), because the training data for foundational models may be, and often is unavailable.

We hope these results answer the question. We remain available throughout the discussion period in case of further questions and discussions. Thank you again for the feedbacks.

References:
- Ilharco et al. Editing models with task arithmetic. ICLR'23

---

### Decision · Program_Chairs · 2024-09-25

**Decision:**

Accept (poster)

**Comment:**

The paper presents an improvement to task arithmetic using different trained coefficients for each block. Overall the reviewers thought that the idea was original, the paper well presented, and the experiments thorough. In the rebuttal, a more thorough discussion of prior work and a multitask finetuning baseline were provided. Otherwise I believe no major weaknesses remain, and I recommend the paper for acceptance. Congratulations to the authors!